# LossVal: Data Valuation using Weighted Loss Functions

## Abstract

Machine learning models are often limited not by how much data we have, but by how much *trustworthy* data we have. We introduce LossVal, a data valuation method that computes per-sample importance scores during neural network training by integrating a self-weighting mechanism into standard loss functions (e.g., cross-entropy and mean squared error). Loss-Val produces meaningful importance scores without repeated retraining and achieves competitive performance on common data valuation tasks such as noisy sample detection and bad point removal. Across multiple classification and regression datasets, LossVal reliably distinguishes helpful from harmful samples. Experiments with ResNet-50 and BERT indicate that LossVal can also be applied to larger architectures in our experimental setup. *Source code and experimental data will be made publicly available upon acceptance.*

## 1 Introduction

In practice, a single mislabeled or atypical training example can affect a model more than thousands of routine samples, motivating methods that quantify the relative importance of individual data points for model optimization, explainability, and data collection (Jia et al., 2021; Molnar, 2022; Chen et al., 2023). This process, known as data valuation, assigns an importance score to each training data point, as illustrated in Figure 1. Applications range from selling or buying data on data markets (Li et al., 2015; Baghcheband et al., 2024) to active learning scenarios where acquiring new, high-impact data is costly (Chen et al., 2023). For example, in passive car safety systems, machine learning models serve as surrogates to predict crash outcomes (Anonymized 3, 2021; Anonymized 2, 2022). Improving these models depends on identifying the most impactful data points, which is challenging due to the presence of both feature and label noise in crash test data. Existing data valuation methods often offer either effective solutions or computational tractability, but not both at once (Park et al., 2023).

We propose a novel data valuation method, *LossVal*, that aims to reduce computational cost relative to retraining-heavy baselines while remaining effective across diverse noise conditions. Our method leverages gradient information from standard loss functions by incorporating learnable parameters into the loss function. By dynamically weighting each data point during training, LossVal identifies beneficial and detrimental data points. We demonstrate that our method performs competitively with strong baselines on six classification and six regression datasets from the OpenDataVal benchmark (Jiang et al., 2023) as well as on the CIFAR-10 (Krizhevsky et al., 2009) and 20Newsgroups (Lang, 1995) datasets. Furthermore, we ~~use~~ evaluate data valuation in an active learning setting to acquire new crash tests: a secondary model is trained on the importance scores to select crash configurations with the highest expected importance, with the goal of improving performance while acquiring as few new data points as possible. ~~This effectively reduces the cost of conducting and acquiring new crash data.~~ Our main contributions are:

- We introduce a self-weighting mechanism for loss functions to compute data importance scores without repeated retraining.

- We achieve strong overall performance across datasets and benchmark tasks, while handling both label and feature noise.

- We assess the use of importance scores to guide data acquisition in costly settings, such as crash tests.

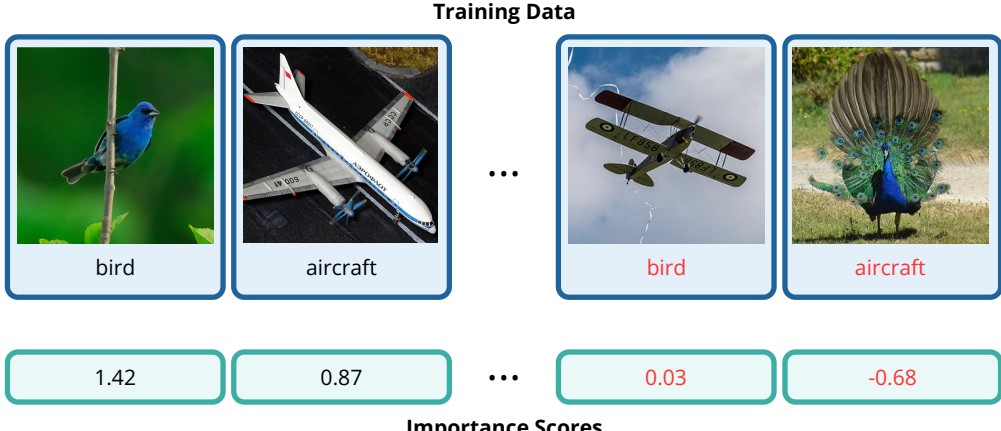

Figure 1: Data valuation methods like LossVal assign a higher importance score to higher-quality data points. Label and feature noise lead to a lower score. Images taken from ImageNet dataset (Deng et al., 2009).

## 2 Related Work

We review relevant literature, focusing on data valuation techniques and machine learning applications in the design of passive car safety systems.

### 2.1 Data Valuation

Data valuation, also known as data attribution (Park et al., 2023), data influence analysis (Hammoudeh & Lowd, 2024), or representer points (Yeh et al., 2018), aims to assign an importance score to each data point in the training data. This score represents how important a data point is for the training of a machine learning model and its performance (Ghorbani & Zou, 2019; Koh & Liang, 2017). There are different approaches to data valuation, each assigning a different meaning or interpretation to the score. Depending on the approach, the importance score is interpreted as influence (Koh & Liang, 2017), Leave-One-Out (LOO) error (Cook, 1977; Bae et al., 2022), Data Shapley value (Ghorbani & Zou, 2019; Chen et al., 2019), or just some form of importance ranking, like the expected utility (Just et al., 2023; Kwon & Zou, 2023; Yoon et al., 2020). Most approaches assume the training set is noisy, the test set is clean, and that one has access to a clean validation set (Just et al., 2023).

We divide data valuation into retraining-based approaches (which include LOO, downsampling, and Shapley value methods), gradient-based approaches, data-based approaches (which focus on the data instead of the model), and other approaches. We discuss representative methods in each category below and reflect on further variants, extensions, and applications of data valuation in Appendix I. Table 1 shows the characterization of some of the most important approaches.

**Retraining-Based Approaches.** LOO (Cook, 1977) is the simplest retraining-based data valuation method. It describes how the model's performance on the test set would change if a given training instance were not included in the training set. It can be calculated by retraining the model $n$ times, leaving out one training instance each time. Some importance scores calculated via LOO may be negative if they lead to a decrease in test performance. As the dataset size increases, the average LOO score decreases. Similarly, Influence-Subsample (Feldman & Zhang, 2020; Jiang et al., 2023) uses subsampled retraining but estimates the same influence values as Influence Functions. Data Shapley profits from the benefits of the Shapley value but is much less efficient (Ghorbani & Zou, 2019; Chen et al., 2019). The Shapley value is a game-theoretic concept for calculating each player's marginal contribution. In data valuation, the training instances are the players working together in a coalition, and the payout is the machine learning model's test performance. When calculating the so-called Data Shapley value, the model needs to be retrained on all possible coalitions of training instances, i.,e., all possible subsets. It profits from the mathematical prop-

Table 1: An overview of a selection of different approaches to data valuation. Volume-based Data Shapley estimates the marginal contribution for different data sources. Shapley Values are optional for DAVINZ. (✓) in the column "Needs clean validation set" means that the method can use a noisy validation set, but performance degrades.

| Method | Shapley Values | No Retraining | Adapts Training | Model-specific | Needs clean validation set |
|---|---|---|---|---|---|
| Leave-One-Out (Cook, 1977) | | | | | ✓ |
| Influence Functions (Koh & Liang, 2017) | | ✓ | ✓ | ✓ | (✓) |
| Representer Point Selection (Yeh et al., 2018) | | ✓ | ✓ | ✓ | (✓) |
| Data Shapley (Ghorbani & Zou, 2019) | ✓ | | | | ✓ |
| Influence-Subsample (Feldman & Zhang, 2020) | | | | | ✓ |
| D-Shapley (Ghorbani et al., 2020) | ✓ | ✓ | ✓ | | ✓ |
| Dropout Influence (Kobayashi et al., 2020) | | ✓ | ✓ | ✓ | |
| TracIn (Pruthi et al., 2020) | | ✓ | ✓ | ✓ | (✓) |
| DVRL (Yoon et al., 2020) | | ✓ | ✓ | | ✓ |
| KNN-Shapley (Jia et al., 2020) | ✓ | ✓ | | ✓ | ✓ |
| KNN-Shapley on Embeddings (Jia et al., 2021) | ✓ | ✓ | | ✓ | ✓ |
| FastIF (Guo et al., 2021) | | ✓ | ✓ | ✓ | (✓) |
| Volume-based Data Shapley (Xu et al., 2021) | (✓) | ✓ | | | |
| Beta Shapley (Kwon & Zou, 2022) | ✓ | | | | |
| AME (Lin et al., 2022) | ✓ | | | | ✓ |
| DAVINZ (Wu et al., 2022) | (✓) | ✓ | ✓ | ✓ | (✓) |
| DU-Shapley (Garrido-Lucero et al., 2023) | ✓ | | | | ✓ |
| LAVA (Just et al., 2023) | | ✓ | | | ✓ |
| Data-OOB (Kwon & Zou, 2023) | | ✓ | | ✓ | |
| TRAK (Park et al., 2023) | | ✓ | ✓ | ✓ | (✓) |
| Data Banzhaf (Wang & Jia, 2023) | | | | | ✓ |
| In-Run DS (Wang et al., 2024a) | ✓ | ✓ | ✓ | ✓ | ✓ |
| Gradient Similarity (Evans et al., 2024) | | | ✓ | ✓ | ✓ |
| **LossVal** | | ✓ | ✓ | ✓ | ✓ |

erties of the Shapley value, additivity (or linearity), efficiency, and symmetry (Shapley, 1951; Ghorbani & Zou, 2019; Molnar, 2023). The time complexity for an exact calculation of the Data Shapley values is in $O(2^n)$ for $n$ training instances (Hammoudeh & Lowd, 2024). Various methods for efficiently approximating the Data Shapley value have been proposed. Average Marginal Effect (AME) (Lin et al., 2022) uses linear regression coefficients to approximate the Shapley values, Beta Shapley (Kwon & Zou, 2022) relaxes the efficiency axiom, DU-Shapley (Garrido-Lucero et al., 2023) draws samples from a uniform distribution, and D-SHAPLEY (Ghorbani et al., 2020) reformulates the Data Shapley to consider the underlying distribution of the data. Kwon & Zou (2023) use bagging to train an ensemble of models on different subsets of the data and estimate importance scores using the Out-of-bag (OOB) error. The importance score of a training instance is the difference between the model performances with and without the instance (Kwon & Zou, 2023).

**Gradient-Based Approaches** utilize training gradients to calculate an importance score. Influence Functions are a staple in statistics for finding influential data points using the Hessian matrix (Cook, 1977; Cook & Weisberg, 1980; Cook, 1986). It can be applied to more complex machine learning tasks (Koh & Liang, 2017; Koh et al., 2019) but is computationally expensive and relies on the convexity of the underlying model (Koh & Liang, 2017). Various techniques have been proposed to approximate the exact values or to speed up the calculation of influence functions (Feldman & Zhang, 2020; Guo et al., 2021; Schioppa et al., 2022). The importance score (or influence) estimated by Influence Functions is more similar to the LOO error or the proximal Bregman response function (Bae et al., 2022). Approaches exploiting gradient information include using the generalized representer theorem to find representer points (Yeh et al., 2018), tracing back gradient updates during training (Pruthi et al., 2020), observing gradient changes in a lower-dimensional space (Park

et al., 2023), and measuring the similarity between training and validation gradients (Evans et al., 2024). Finally, gradient information is also used to estimate Data Shapley values with a single training run (Wang et al., 2024a; Cai, 2024). *LossVal also falls into the gradient-based category. However, we exploit the gradient information only implicitly during the model training.*

**Data-Based Approaches.** So far, the approaches described relied on machine learning models to estimate an importance score, i. e., the score is biased towards the model. Model-agnostic approaches assign importance scores to data points based only on the data (Xu et al., 2021). Xu et al. (2021) calculate a volume measure for each data point by considering the diversity of the data, which correlates with learning performance. Just et al. (2023) optimize a weighted optimal transport distance to calculate the distance between noisy training data and clean validation data, interpreting it as a proxy for test performance.

**Other Approaches.** Not all approaches fit the previous three categories. Kobayashi et al. (2020) identify sub-networks of a neural network that were trained slightly differently, resulting from dropout zero-masking, and analyze how different sub-networks perform based on their training data. DAVINZ (Wu et al., 2022) uses the generalization boundary to estimate how a change in the training data would change the test performance. DVRL applies reinforcement learning to estimate importance scores (Yoon et al., 2020). Various approaches use $k$-Nearest-Neighbors (KNN) methods to estimate Data Shapley values, as these can be calculated more efficiently with KNN (Jia et al., 2020; 2021; Belaid et al., 2023).

## 2.2 Machine Learning in Passive Car Safety

In passive car safety, the focus is on systems that protect vehicle occupants during a crash, such as airbags, belt force limiters, and irreversible belt pretensioners. Unlike active safety systems, which aim to prevent collisions (e. g., lane-keeping assistance or emergency braking systems), passive safety features are only triggered once a collision is unavoidable. Ethical concerns surrounding interventions that actively control the vehicle (e. g., swerving into oncoming traffic) (Hansson et al., 2021) do not apply to passive safety systems, as their purpose is purely protective.

Machine learning techniques have been applied to optimize various parameters of passive safety systems (Anonymized 3, 2021; Liu et al., 2023; Mathieu et al., 2024; Anonymized 2, 2022; Sun et al., 2023; Anonymized 1, 2024). Key optimizations include the belt force load limiter, airbag vent hole size, and load limiter level switching time. These optimizations are critical for minimizing injury risk during collisions (Anonymized 1, 2024). For example, Anonymized 3 (2021) employed a convolutional neural network to predict chest acceleration during a crash based on vehicle and restraint system parameters. Similarly, Anonymized 2 (2022) introduced the Real Occupant Load Criterion ($ROLC_p$), a metric used to estimate crash severity. Their approach used a combination of machine learning models to predict the $ROLC_p$ from vehicle data and restraint system configurations. In another study, Mathieu et al. (2024) applied reinforcement learning to determine restraint system parameters that yielded lower occupant loads than traditional methods. Furthermore, Sun et al. (2023) demonstrated that Gaussian processes can be used to dynamically adjust restraint system parameters based on the occupant's height and weight, further improving occupant protection in real-world accidents.

# 3 LossVal

~~With LossVal~~In this section, we introduce ~~instance-specific weights into the loss function to estimate and update~~ LossVal, a novel approach to estimating the importance of ~~samples during training~~ training samples by integrating learnable weights into the loss function. The instance-specific weights are part of the loss function but updated using gradient descent during model training.

**Optimization of instance weights.** ~~Let $\theta$ denote the model parameters and let $w \in \mathbb{R}^N$ denote the instance-specific weights for the $N$ training instances. During training, we optimize both $\theta$ and $w$ by gradient~~

~~descent (potentially with different learning rates $\eta_\theta$ and $\eta_w$):~~

$$\theta^{(t+1)} = \theta^{(t)} - \eta_\theta \, \nabla_\theta \, \text{LossVal}(\theta^{(t)}, w^{(t)}),$$

$$w^{(t+1)} = w^{(t)} - \eta_w \, \nabla_w \, \text{LossVal}(\theta^{(t)}, w^{(t)}).$$

~~We interpret the final (or epoch-averaged) weights $w$ after training as the importance scores; larger values indicate more beneficial samples under the chosen target loss.~~

The proposed loss function ~~LossVal~~ has two factors, an instance-weighted target loss $\mathcal{L}_w$ (e.g., weighted cross-entropy loss, weighted MSE) and the optimal transport distance $\text{OT}_w$. Let $w \in \mathbb{R}^N$ denote the instance-specific weights for the $N$ training instances:

$$\text{LossVal} = \mathcal{L}_w(y, \hat{y}) \cdot \text{OT}_w(X_{train}, X_{val})^2. \tag{1}$$

We treat the instance weights $w$ as learnable parameters and interpret their optimized values after training as importance scores. For the target loss ~~$\mathcal{L}w$~~$\mathcal{L}_w$, we use instance-weighted formulations of existing loss functions, like a weighted cross-entropy loss or weighted mean-squared error. ~~Here, $w$ represents the learnable importance scores, i.,e, the weights assigned to each data point. Each data point is weighted separately.~~ The model's prediction is denoted by $\hat{y}$, while $y$ represents the target values. The optimal transport distance $\text{OT}_w$ takes the features of the training data $X_{train}$ and validation data $X_{val}$ as input.

~~The weighted formulations of loss functions add learnable weights to the local loss, one per training instance.~~ All weights $w_n$ are initialized to 1. In practice, we interpret $w$ mainly in terms of relative importance~~; if desired, $w$ can be normalized (e.g., to have mean 1) without changing the induced ranking~~. The model learns to down-weight noisy or less informative data points and up-weight highly informative ones. Incorporating the weighted distribution distance $\text{OT}_w$ ensures that the feature space is also considered when optimizing the instance-specific weights. We square the optimal transport term (i.e., use $\text{OT}_w^2$) because it performs better empirically than the unsquared distance; we confirm this in the ablation study (Section 6).

We multiply $\mathcal{L}_w$ and $\text{OT}_w$ in Eq. 1 (rather than adding them) for two reasons. ~~First, the~~ The product is zero ~~once the~~ only when the global target loss $\mathcal{L}_w$ ~~is zero, so the instance-specific weights stop changing after the model has fit the targets for an instance~~ vanishes. For an individual well-fit instance, the residual term drops out but the transport term $\mathcal{L}_w \sum_j c(x_i, x_j)$ persists, so already-fit points continue to be reweighted according to feature-space alignment rather than freezing. Second, the product yields a clearer learning signal for the weights: the gradient of LossVal with respect to $w_j$ is scaled by the other factor, which couples the updates across instances and makes the weights $w_i$ depend on the remaining weights $w_j$ with $j \neq i$ (see Appendix A). ~~Finally, we square the optimal transport term (i.e., use $\text{OT}^2$) because it performs better empirically than the unsquared distance; we confirm this in~~ Intuitively, the ~~ablation study ().~~ three components play complementary roles: the target loss $\mathcal{L}_w$ adjusts each weight according to how well the label is predicted, the transport term $\text{OT}_w$ adjusts it according to how close the point lies to the validation distribution in feature space, and the product couples the two so that a point retains high weight only if it is both well-explained and distributionally in-domain.

**Optimization of instance weights.** During training, we optimize both $\theta$ and $w$ by gradient descent (potentially with different learning rates $\eta_\theta$ and $\eta_w$). Let $\theta$ denote the model parameters:

$$\theta^{(t+1)} = \theta^{(t)} - \eta_\theta \, \nabla_\theta \, \text{LossVal}(\theta^{(t)}, w^{(t)}), \tag{2}$$

$$w^{(t+1)} = w^{(t)} - \eta_w \, \nabla_w \, \text{LossVal}(\theta^{(t)}, w^{(t)}). \tag{3}$$

We interpret the final weights $w$ after training as the importance scores; larger values indicate more beneficial samples under the chosen target loss.

**Weighted Loss for Classification.** The cross-entropy loss (CE) is widely used for classification tasks (Wang et al., 2022). We incorporate the data valuation into ~~CE~~ CE by introducing instance-specific

weights $w_n$:

$$\mathrm{CE}_w = -\sum_{n=1}^{N} \left[ w_n \cdot \sum_{k=1}^{K} y_{n,k} \log(\hat{y}_{n,k}) \right], \quad \mathrm{LossVal}_{\mathrm{CE}} = \mathrm{CE}_w \cdot \mathrm{OT}_w(X_{train}, X_{val})^2, \tag{4}$$

where $N$ denotes the number of training samples, $K$ denotes the number of classes in the training set, $y_{n,k}$ the true class vector, $\hat{y}_{n,k}$ is the prediction of the model, and $w_1, \ldots, w_N$ are the instance-specific weights (which are interpreted as the importance scores).

Two key points distinguish LossVal from other weighted loss functions. First, the weights are applied per instance rather than per class, as in focal loss (Lin et al., 2018). Second, our weights are learnable parameters optimized during training via gradient descent. This bears similarities to self-paced learning (Kumar et al., 2010), which dynamically adjusts the subset of training samples for fitting based on their difficulty.

**Weighted Loss for Regression.** For regression, the mean squared error (MSE) is widely used (Wang et al., 2022). We incorporate the instance weights into the MSE similarly to the modification of the cross-entropy loss, with $N$ samples, target value $y_n$, predicted value $\hat{y}_n$, and instance-specific weights $w_n$.

$$\mathrm{MSE}_w = \sum_{n=1}^{N} w_n \cdot (y_n - \hat{y}_n)^2, \quad \mathrm{LossVal}_{\mathrm{MSE}} = \mathrm{MSE}_w \cdot \mathrm{OT}_w(X_{train}, X_{val})^2. \tag{5}$$

Similarly, iteratively reweighted least squares (Holland & Welsch, 1977) is a linear regression technique that dynamically adjusts instance-based weights during optimization. It primarily aims to downweight outliers to improve model fit, which differs from the objectives of LossVal. Furthermore, LossVal performs more complex gradient computations by integrating the optimal transport distance into the MSE.

**Weighted Optimal Transport.** The Sinkhorn distance is an entropically regularized approximation to the Wasserstein distance, which measures the optimal transport cost between two distributions (Cuturi, 2013). The Sinkhorn distance can be calculated between discrete distributions of different sizes while remaining differentiable (Feydy et al., 2019). By introducing instance-based weights into the Sinkhorn distance, we optimize not only the target loss but also the distributional alignment between training and validation data. The target loss $\mathcal{L}_w$, e.g., a modified cross-entropy loss or modified MSE, mainly considers the labels and the models' prediction. This means that a weighted target loss mostly adapts the weights based on information in labels and predictions. We incorporate the distribution of the input features of the data points into the loss by multiplying $\mathcal{L}_w$ with a weighted optimal transport distance $\mathrm{OT}_w$ (see Eq. 1), allowing feature information to guide the optimization of instance-specific weights, too. The optimal transport cost between two distributions ($X_{train}$ and $X_{val}$) is defined as the ~~fastest~~ least-cost way to move all points from the source to the target distribution (Cuturi, 2013).

$$\mathrm{OT}_w(X_{train}, X_{val}) = \min_{\gamma \in \Pi(w,1)} \sum_{n=1}^{N} \sum_{j=1}^{J} c(x_n, x_j)\, \gamma_{n,j}, \tag{6}$$

where $\Pi(w, 1)$ is the set of all joint probability distributions $\gamma$ with marginal $w$ for the training set and a uniform marginal for the validation set, ensuring the transport plan respects the instance-specific weights $w$. Each $\gamma$ defines a possible transport plan for moving the training distribution to the validation distribution. The optimal transport plan $\gamma*$ is the transport plan that leads to the shortest distance. The cost function $c(x_n, x_j)$ denotes the effort of transport, typically the squared Euclidean distance $\|x_n - x_j\|^2$. $N$ is the number of training data points, and $J$ is the number of validation data points.

Sinkhorn's distance adds the entropy $H(\gamma)$ as a regularization term to OT, which makes OT differentiable and typically yields a more tractable approximation than solving the exact optimal transport problem (Cuturi, 2013; Feydy et al., 2019). Just et al. (2023) showed that Sinkhorn's distance can be effectively utilized in the data valuation context, but it would be possible to use any other weighted distributional distance, too. By including the weighted $\mathrm{OT}_w$ in the loss function, gradient descent optimizes the weights to decrease the optimal transport distance between the training and validation sets, effectively reducing the importance

scores of training points that differ from the validation distribution. Training data points that are more similar (i,e., closer) to the data points in the validation set get up-weighted, while more different data points get down-weighted.

## 4 Experimental Apparatus

**Datasets.** We employ six widely used classification datasets, which are the focus of the OpenDataVal benchmark (Jiang et al., 2023). OpenDataVal does not include predefined regression datasets, so we select six datasets from the CTR-23 benchmark suite (Fischer, 2023). To demonstrate that LossVal can be applied effectively to larger-scale datasets and models, we use CIFAR-10 (Krizhevsky et al., 2009) and 20Newsgroups (Lang, 1995) datasets. Additionally, we employ a crash test dataset consisting of $1,122$ samples from the NHTSA database (National Highway Traffic Safety Administration (NHTSA), 2024) and 154 proprietary crash tests provided by a large car manufacturer (Anonymized 3, 2021; Anonymized 2, 2022) to evaluate LossVal in an active data acquisition setting. Details of the datasets are provided in Appendix D.

**Procedure.** We compare LossVal against 10 baselines that span different approaches to data valuation. These are Data Shapley (Ghorbani & Zou, 2019; Chen et al., 2019), Beta Shapley (Kwon & Zou, 2022), Leave-One-Out (Cook, 1977), KNN-Shapley (Jia et al., 2020), Data Banzhaf (Wang & Jia, 2023), AME (Lin et al., 2022), Influence Subsample (Feldman & Zhang, 2020), LAVA (Just et al., 2023), DVRL (Yoon et al., 2020), and Data-OOB (Kwon & Zou, 2023). The baselines are selected based on the OpenDataVal benchmark (Jiang et al., 2023). We compare LossVal and the baselines on the tasks defined in the OpenDataVal benchmark (Jiang et al., 2023), including Noisy Label Detection, Noisy Feature Detection, Mixed Noise Detection, Point Addition, and Point Removal.

Many existing data valuation methods rely on repeated model training. We limit the number of training epochs to ensure a fair comparison between LossVal and the baselines, and evaluate LossVal with 5 and 30 epochs. LossVal with 5 epochs demonstrates how it performs when trained for the same number of epochs as each model of the baseline methods. LossVal with 30 epochs is a fairer comparison to methods like Data-OOB or LOO that train $1,000$ models for $5,000$ epochs overall. We repeat every experiment 15 times.

For CIFAR-10 and 20Newsgroups we use ResNet-50 (He et al., 2015) and (Devlin et al., 2019), respectively. Because of the high complexity of those models, most of the baselines above are not realistically applicable. For example, applying Data-OOB to CIFAR-10 would mean retraining the ResNet model $45,000$ times from scratch. To resolve this, we use LAVA (Just et al., 2023) and KNN-Shapley (Jia et al., 2020) from the baselines above, as they do not require training a model. DataInf (Kwon et al., 2024) is added as a fast approximation of Influence Functions. It replaces the slower InfluenceSubsample, which is also an approximation of Influence Functions. This way, we cover comparisons with Data Shapley-based, Influence-based, and model-free methods. The experiments on CIFAR-10 and 20Newsgroups are repeated 5 times. Finally, we demonstrate LossVal's effectiveness for active data acquisition using a crash test dataset.

*Noisy Sample Detection Tasks.* We introduce label noise (where $p\%$ of the labels get mixed), feature noise (add Gaussian noise to $p\%$ of samples), or both into $p\%$ of the labels, where $p \in \{5, 10, 15, 20\}$. We evaluate how well each data valuation method detects noisy points. Noisy samples often contain errors or irrelevant information that can mislead the learning algorithm, reducing the model's performance. An effective data valuation method should assign lower importance scores to these noisy samples. To assess robustness under varying data corruption types, we consider three noise regimes: noisy labels (where $p\%$ of the labels are randomly permuted), noisy features (adding Gaussian noise to $p\%$ of samples), and mixed noise ($\frac{p}{2}\%$ label noise and $\frac{p}{2}\%$ feature noise).

*Point Removal and Point Addition Tasks.* We test how removing the most valued data points from the training set affects the model performance. Removing valued data points should cause model performance to degrade faster than random removal. We start with the complete training set and iteratively remove the $5\%$ top-valued points, from $0\%$ to $50\%$ of the points, retraining each time. We use $20\%$ noise on the training samples (either noisy labels, noisy features, or mixed noise) and evaluate test performance with logistic and linear regression models, as these simpler models are less prone to overfitting when the dataset is very small. The point addition task starts with $5\%$ training data. Then, $5\%$ of the least-valued data points are added

iteratively until we reach 50%. The performance of a good data valuation method should increase more slowly than randomly adding data points.

*Experiments with Larger Models.* The longer training time and larger dataset sizes of the larger scale experiments force us to limit our experiments to an informative minimum. Due to resource constraints, we limit experiments with those models to noisy sample detection with 20% noisy labels. Each experiment is repeated 5 times.

*Active Data Acquisition Task.* The regression crash test dataset is sorted by time, and the first 40% is used for training. The rest of the data points are randomly allocated to 10% validation, 40% acquisition, and 10% test data. This emulates the process of acquiring new data, where we only add crash tests from newer car models to the training set.

Due to the potential for injuries from sub-optimally designed restraint systems and the high costs of conducting new crash tests, there is substantial interest in adding only high-quality data points to the training data and minimizing the number of data points required to improve the performance of the machine learning model.

First, a crash model is trained to predict the severity of a crash on an occupant. A secondary model is employed to guide the active data acquisition process by estimating the potential improvement in the crash model's performance when adding new data points. Details on the procedure and two models involved are described in Appendix E.

**Hyperparameter Optimization.** We use three different MLP models: one for classification tasks, one for regression tasks, and one for active data acquisition using crash data. Using grid search, we optimized hyperparameters to maximize accuracy and the $R^2$ score on the target variable. The hyperparameters are described in Appendix F. For CIFAR-10 (Krizhevsky et al., 2009), we use ResNet-50 (He et al., 2015), and for 20Newsgroups, we use BERT instead. The Adam optimizer (Kingma & Ba, 2017) is used for all experiments. For the baseline methods, we used the hyperparameters provided by the OpenDataVal benchmark (Jiang et al., 2023). For LossVal, we finetuned the learning rate separately ~~and~~ to account for the different loss function which affects the learning dynamics. We found that 0.01 was best for both tasks. The same learning rate is used for the MLP parameters and the weights $w_n$ in the loss function to keep the setup simple and to account for the fact that tuning the learning rate separately may not be possible in a less controlled environment.

**Measures.** For the *Noisy Sample Detection Tasks*, we report the noisy sample detection curves and F1-scores for all methods, averaged across all datasets and runs. The noisy sample detection curves show the proportion of noisy samples detected by inspecting data points with low importance scores. For better comparability, we also report the average of the curve. Further, the balanced F1 score is calculated to see how many of the actual noisy samples the data valuation detected. The F1 score is reported for each data valuation method at each noise level. Additionally, we report the overall average F1 score per method. For the evaluation with ResNet-50 and BERT, we only report the F1 scores for 20% noisy samples.

For the *Point Addition and Point Removal Tasks*, we present the test performance curve of removing or adding the most or least valued data points, respectively. Lower curves indicate better data valuation. For better comparability, we also report the average of the curve.

Regarding the *Active Data Acquisition*, we measure the change resulting from adding 1% additional data points and retrain the model on the updated dataset. Then we compare the MSE, $R^2$-score, and mean absolute percentage error (MAPE) of the original model on the test set with the updated model.

## 5 Results

**Noisy Sample Detection.** Figure 2 shows how well each data valuation method can find noisy data points. The x-axis describes the ratio of noisy samples, and the y-axis describes how well the method performs. The plots are divided into learning tasks (regression vs. classification) and noise types. We observe that no single data valuation performs the best for all tasks and noise types. LAVA (Just et al., 2023) outperforms the others in the detection of noisy features. Data-OOB (Kwon & Zou, 2023) performs well in detecting noisy

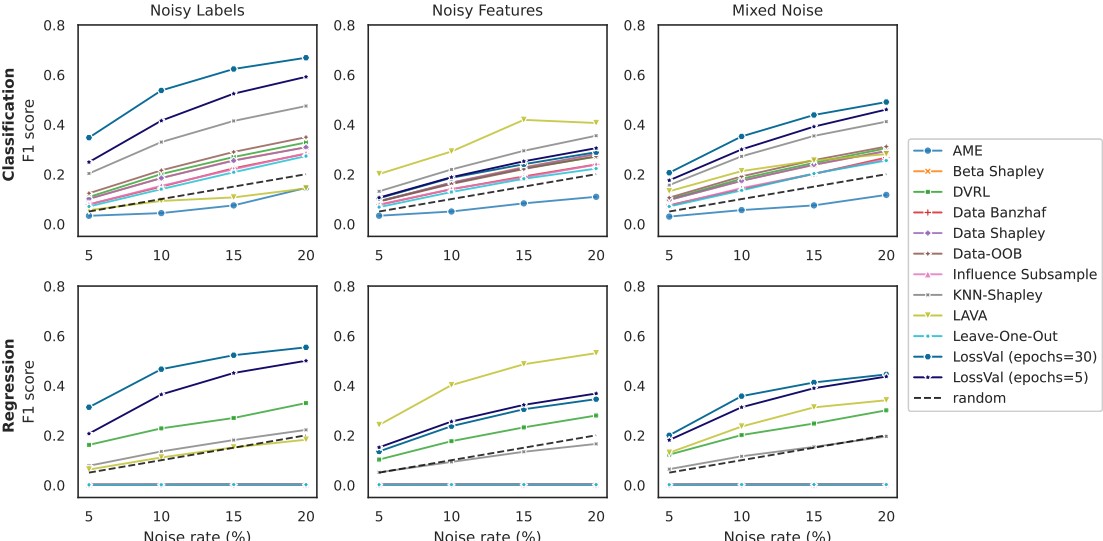

Figure 2: F1 scores calculated between the set of correct noisy samples and the noisy samples found and averaged over all datasets. Higher is better.

labels in classification tasks but struggles with noisy features and regression. KNN-Shapley (Jia et al., 2020) shows strong performance in both noisy label and noisy feature detection for classification. DVRL (Yoon et al., 2020) shows good performance for both regression and classification, but is outperformed by other methods in every task. LossVal performs well on noisy-label and mixed-noise detection, even outperforming all other methods for both regression and classification.

Table 2 shows the average F1 score over all noise levels. LossVal, KNN-Shapley, and LAVA show the best performance in classification tasks. In regression tasks, LossVal outperforms all other methods for mixed-noise and noisy-label detection but is second to LAVA in noisy-feature detection. It is notable that LossVal gets worse in detecting label noise after longer training. Feature noise is more nuanced than label noise and after training the model for longer, it might learn to handle the feature noise better, thereby weakening the signal for the coefficients in the loss function. The difference is small and could also be due to random chance. Section G.1 details how well the different approaches can detect noisy samples.

**Point Addition and Removal.** Figure 3 shows the effect of adding the data points with the lowest importance score to the training set and then retraining the MLP on the updated training set. For regression, we normalized all values per dataset before averaging over all datasets. Lower curves are better because they indicate a slower increase in test performance when low-importance data points are added to the training set. For classification, KNN-Shapley performs the best, LAVA comes in second, and DVRL third. For regression, DVRL performed best, followed by LAVA and KNN-Shapley. We provide the numerical values in Section G.2.

The point removal experiment starts from the reverse premise: Removing data points with a high importance score should lead to a steeper decrease in test performance than randomly removing data points. Data valuation methods that lead to a lower point-removal curve, ~~as~~ shown in Figure 4, are better at identifying high-quality data points. In classification, KNN-Shapley achieves the best score, followed by LossVal and DVRL. For regression, DVRL performs best, LossVal second-best, and LAVA and KNN-Shapley tie for third-best, achieving similar results. The plots show that LossVal performs worse than other methods after removing just a few points, but catches up to the best methods after removing more points.

~~In summary, LossVal and KNN-Shapley outperform all other methods in the point removal experiments. LossVal is better at finding high-quality data points in classification tasks, but KNN-Shapley achieves much better results than all other methods on regression tasks.~~

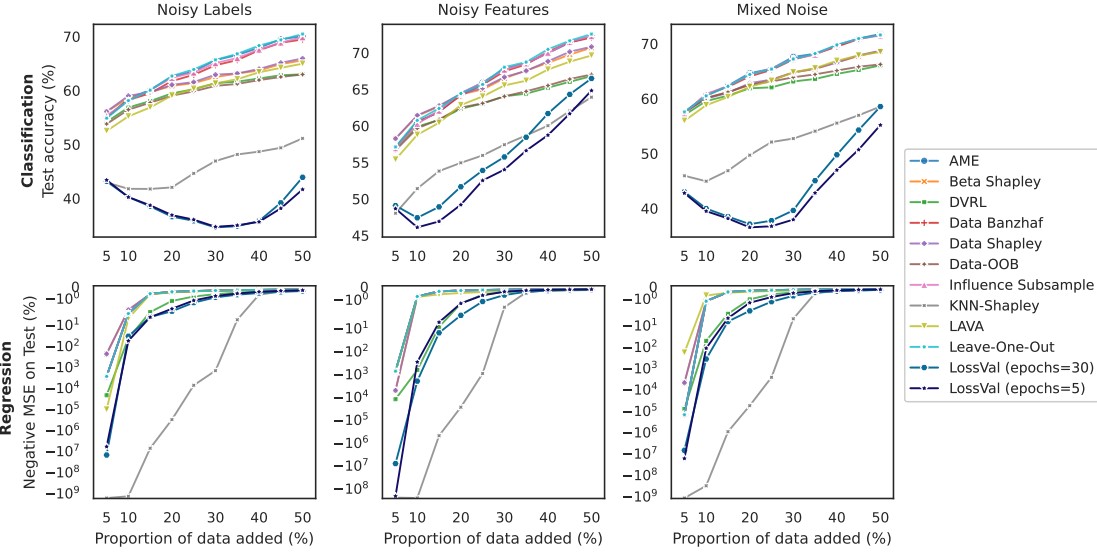

Figure 3: Adding $x\%$ of data points with low importance score to the training data, averaged over all datasets. A lower curve is better.

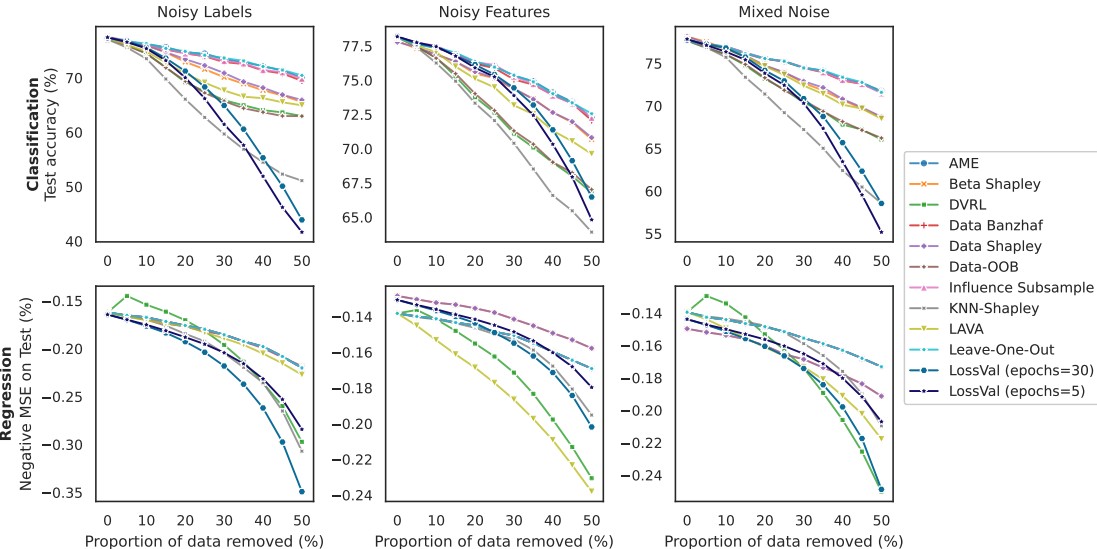

Figure 4: Removing $x\%$ data points with a high importance score from the training data. A lower curve is better.

Table 2: Average of the noisy sample detection F1 scores of each data valuation method, averaged over all noise rates and datasets. The number after ± indicates the standard error. Higher is better.

| | | Noisy Labels | Noisy Features | Mixed Noise | Overall Average |
|---|---|---|---|---|---|
| **Classification** | AME | 0.074±.005 | 0.069±.005 | 0.069±.005 | 0.071±.003 |
| | Beta Shapley | 0.212±.003 | 0.191±.003 | 0.198±.003 | 0.201±.002 |
| | DVRL | 0.226±.005 | 0.187±.003 | 0.208±.004 | 0.207±.002 |
| | Data Banzhaf | 0.184±.004 | 0.162±.004 | 0.171±.004 | 0.172±.002 |
| | Data-OOB | 0.244±.005 | 0.186±.003 | 0.216±.003 | 0.215±.002 |
| | Data Shapley | 0.212±.003 | 0.191±.003 | 0.198±.003 | 0.200±.002 |
| | Influence Subsample | 0.184±.004 | 0.161±.004 | 0.170±.004 | 0.171±.002 |
| | KNN-Shapley | *0.355*±.006 | 0.250±.005 | *0.298*±.005 | *0.301*±.003 |
| | LAVA | 0.099±.004 | **0.329**±.012 | 0.220±.008 | 0.216±.005 |
| | Leave-One-Out | 0.173±.004 | 0.150±.004 | 0.166±.004 | 0.163±.003 |
| | LossVal (epochs=5) | 0.445±.007 | *0.213*±.003 | 0.332±.005 | 0.330±.004 |
| | LossVal (epochs=30) | **0.544**±.008 | 0.204±.004 | **0.371**±.005 | **0.373**±.005 |
| **Regression** | AME | 0.002±.000 | 0.002±.000 | 0.002±.000 | 0.002±.000 |
| | Beta Shapley | 0.002±.000 | 0.002±.000 | 0.002±.000 | 0.002±.000 |
| | DVRL | *0.247*±.007 | 0.198±.004 | 0.218±.005 | 0.221±.003 |
| | Data Banzhaf | 0.002±.000 | 0.002±.000 | 0.002±.000 | 0.002±.000 |
| | Data-OOB | 0.002±.000 | 0.002±.000 | 0.002±.000 | 0.002±.000 |
| | Data Shapley | 0.002±.000 | 0.002±.000 | 0.002±.000 | 0.002±.000 |
| | Influence Subsample | 0.002±.000 | 0.002±.000 | 0.002±.000 | 0.002±.000 |
| | KNN-Shapley | 0.154±.008 | 0.111±.005 | 0.132±.006 | 0.132±.004 |
| | LAVA | 0.127±.004 | **0.415**±.012 | *0.255*±.008 | *0.265*±.006 |
| | Leave-One-Out | 0.002±.000 | 0.002±.000 | 0.002±.000 | 0.002±.000 |
| | LossVal (epochs=5) | 0.380±.007 | 0.274±.005 | 0.330±.006 | 0.328±.004 |
| | LossVal (epochs=30) | **0.464**±.008 | *0.256*±.006 | **0.354**±.006 | **0.358**±.004 |

Table 3: F1 scores calculated between the set of correct noisy samples and the noisy samples found. The number after ± indicates the standard error. Higher is better.

| | 20Newsgroups | CIFAR-10 |
|---|---|---|
| KNN-Shapley | 0.324±0.005 | **0.41**±0.004 |
| KNN-Shapley (fine-tuned) | 0.615±0.013 | — |
| LAVA | 0.229±0.004 | 0.217±0.004 |
| LAVA (fine-tuned) | 0.662±0.028 | — |
| DataInf | 0.264±0.016 | 0.332±0.028 |
| LossVal | **0.756**±0.005 | 0.36±0.005 |

**Experiments with Larger Models.** LossVal is competitive on both the CIFAR-10 and 20Newsgroups dataset (compare Table 3). It outperforms all other methods on the 20Newsgroups dataset with BERT and ranks second-best on CIFAR-10 with ResNet-50. While KNN-Shapley achieves a higher score on CIFAR-10, it is also roughly 3 times slower than LossVal. The runtime comparison in Appendix H shows that LossVal is as efficient as the fastest baselines and significantly faster than DataInf, which also relies on the training of a model.

**Active Data Acquisition.** There are no strong differences between the data valuation methods in the active data acquisition experiment. As shown in Table 8 in Section G.3, AME, Beta Shapley, and Data Shapley improve the $MSE$ the most. AME achieves the best $R^2$-score, with Beta Shapley coming second and Data-OOB as third.

# 6 Ablations

We perform ablations on LossVal components to demonstrate their importance. Further, we investigate how LossVal affects the downstream classification and regression performance.

**Importance of LossVal Components.** Table 4 indicates how the results for LossVal change if parts of the loss function are left out. We see that all parts of LossVal are important for the results. Furthermore, the multiplication of target loss and distribution distance cannot be replaced by addition.

Table 4: Ablation study showing the effect of removing parts of the LossVal loss function on the noisy sample detection. The number after ± indicates the standard error. Higher is better.

|  |  | Noisy Labels | Noisy Features | Mixed Noise | Overall Average |
|---|---|---|---|---|---|
| Classification | $OT_w$ | 0.146±.003 | 0.214±.004 | 0.196±.003 | 0.185±.002 |
|  | $OT_w^2$ | 0.133±.003 | 0.160±.003 | 0.157±.003 | 0.150±.002 |
|  | $CE_w$ | **0.546**±.007 | 0.153±.004 | 0.367±.005 | 0.356±.005 |
|  | $CE_w + OT_w$ | 0.159±.003 | **0.216**±.004 | 0.201±.003 | 0.192±.002 |
|  | $CE_w + OT_w^2$ | 0.115±.003 | 0.110±.002 | 0.115±.003 | 0.113±.001 |
|  | $CE_w \cdot OT_w$ | *0.388*±.005 | 0.196±.004 | *0.298*±.004 | *0.294*±.003 |
|  | LossVal | 0.544±.008 | *0.204*±.004 | **0.371**±.005 | **0.373**±.005 |
| Regression | $OT_w$ | 0.117±.003 | 0.137±.003 | 0.134±.003 | 0.129±.002 |
|  | $OT_w^2$ | 0.137±.003 | 0.253±.005 | *0.217*±.004 | 0.202±.002 |
|  | $MSE_w$ | *0.230*±.005 | 0.124±.003 | 0.184±.003 | 0.179±.002 |
|  | $MSE_w + OT_w$ | 0.142±.003 | *0.252*±.005 | 0.217±.004 | *0.203*±.002 |
|  | $MSE_w + OT_w^2$ | 0.099±.002 | 0.099±.002 | 0.099±.002 | 0.099±.001 |
|  | $MSE_w \cdot OT_w$ | 0.395±.007 | 0.213±.005 | 0.304±.005 | 0.304±.004 |
|  | LossVal | **0.464**±.008 | **0.256**±.006 | **0.354**±.006 | **0.358**±.004 |

**Effect of LossVal on Performance.** Using LossVal during training changes the loss function and, therefore, the gradients used to update the model parameters. To better understand how much the loss modification affects the model's test performance, we compared MLPs trained with or without LossVal. We trained an MLP with the same hyperparameters as in previous experiments on both the regression and classification benchmarks. We repeated the training 15 times per dataset with a standard target loss (MSE or cross-entropy loss) and the respective LossVal loss. We calculate classification accuracy and $R^2$ for regression datasets.

We found no strong difference between the test performance of a model trained using a standard loss or using a LossVal loss. For classification, we reject the hypothesis that using LossVal reduces the test accuracy compared to using the cross-entropy loss, as no statistically significant difference was found between the two conditions, $t(178) = -0.005$, $p = 0.995$. For regression, we fail to reject the hypothesis that LossVal reduces the test $R^2$ score relative to using the MSE, $t(178) = 1.350$, $p = 0.179$.

## 7 Discussion

We have compared LossVal to a range of methods covering all branches of data valuation identified in Section 2. The comparison with 10 baseline methods across 13 datasets provides a comprehensive picture of LossVal's performance. Our experiments demonstrate that LossVal ~~matches or outperforms~~ is competitive with state-of-the-art data valuation methods on the OpenDataVal benchmark tasks on both small and large datasets. LossVal's performance is ~~robust across~~ more robust to different types of noise ~~and for~~ than the baselines and achieves similar performance on both regression and classification tasks~~, and it successfully identifies beneficial and detrimental data points during~~. This is a clear advantage to previous data valuation approaches which performance varies strongly depending on the type of noise and the task.

Using data valuation for active data acquisition ~~tasks.~~ only leads to slightly better results than randomly selecting the parameters for the next crash test. This possibly stems from the inductivity of our approach: We use the importance of *known* data points to estimate the importance of *unknown* data points. Data points that are very different from the data points in the training set potentially highly impact the generalization of the model. However, the expected importance of very different data points underlies a higher uncertainty. Basically, the secondary model can only accurately predict the importance scores for data points that are similar to the training data points, but data points that are very different may have a higher impact on the test performance.

The algorithmic complexity of LossVal is ~~in $O(n + T)$~~ roughly in $O(T + e \cdot (n/b) \cdot (b \cdot v))$, where $n$ is the dataset size ~~and~~, $T$ represents the complexity of a single training run~~.~~, $b$ is the batch size, and $v$ is the size of the validation set. Note that the complexity if approximately quadratic based on the size of the compared sets, but is constraint in our application as it is only applied batch-wise. Assuming a fixed batch size and fixed validation set size, the overhead created by LossVal increases linearly with respect to the number of training samples. This is a lower complexity than other data valuation approaches. Retraining-based methods like Leave-One-Out (LOO) exhibit a time complexity of $O(n \cdot T)$, making them impractical for large datasets due to the repeated retraining (Hammoudeh & Lowd, 2024). While it has useful theoretical properties, the exact calculation of Data Shapley is in $O(2^n \cdot T)$, making it untractable in most situations. Gradient-tracking methods such as TracIn (Pruthi et al., 2020) have a time complexity of $O(n \cdot p)$, where $p$ is the number of model parameters, because they require constant gradient tracking across iterations, which adds computational overhead (Hammoudeh & Lowd, 2024). Influence-based approaches like Influence Functions achieve an $O(n \cdot p)$ complexity by leveraging Hessian approximations (Hammoudeh & Lowd, 2024). Runtimes are reported in Appendix H. We use a relatively inefficient but well-tested and robust implementation of Sinkhorn's distance. More efficient implementations may improve LossVal's runtime (Just et al., 2023). Still, LossVal achieved a lower runtime than all other approaches that are based on model training. Methods like KNN-Shapley are faster, as they do not even train a model. However, the importance values are not specific for a model and can therefore not help in explaining or interpreting a model's behavior.

~~We have compared LossVal to a range of methods covering all branches of data valuation identified in . The comparison with 10 baseline methods across 13 datasets provides a comprehensive picture of LossVal's performance.~~

The importance scores generated by LossVal are less informative than those generated by some other methods. The LOO and Shapley values quantify whether and how much a model's test performance improves or decreases if a data point is removed. The importance scores of LossVal cannot express this, but an exact quantification is not necessary for most applications.

Although Data-OOB is model-agnostic, we observed that it performs better with logistic regression as the base model than with an MLP (see Section G.5 for details). To ensure a fair comparison across all data valuation methods, we avoided tuning model hyperparameters individually per method.

# 8 Conclusion and Future Work

LossVal is an effective data valuation method for neural networks. It achieves state-of-the-art results and consistently outperforms the state of the art in regression while requiring only a single training run. ~~Unlike many~~ Compared to most existing data valuation methods, LossVal maintains ~~robust~~ a good performance regardless of the noise type or task.

Directions for further exploration include investigating whether LossVal can be successfully extended to different loss functions, such as hinge loss, focal loss, or others. An especially interesting avenue is the extension of LossVal to LLMs, which use cross entropy at multiple token position in parallel, which requires a special adaption before LossVal can be applied. Furthermore, it is challenging to compare results across different methods due to inconsistencies in benchmarks and reporting. Establishing a standard score for data valuation methods would benefit the field and allow meaningful comparisons.

**Broader Impact Statement**

~~This paper aims to enhance the understanding of how individual data points influence machine learning model training. By applying the LossVal method to crash test data, we demonstrate its significant utility in the Automotive Passive Safety domain. This approach not only aids in refining machine learning pipelines but also unveils previously undiscovered patterns that complement mechanical engineering efforts, ultimately contributing to the development of safer roads. While our work has broad implications, particularly in improving data quality and data-driven decision-making, we believe that no specific societal consequences require immediate emphasis in this context.~~

Data valuation methods like LossVal can improve data quality assessment and data-driven decision-making. Our crash test application demonstrates feasibility in automotive passive safety, though guided acquisition did not substantially outperform random selection in our experiments (Table 8), so we make no claim of immediate practical safety benefit. One risk warrants emphasis: LossVal down-weights training points far from the validation distribution (Section 3), which in safety-critical settings could deprioritize rare but important configurations. Importance scores should therefore complement, not replace, expert judgment. The dataset includes 154 proprietary crash tests used under a data usage agreement with the manufacturer and cannot be released, the majority of the crash data is publicly available at the NHTSA database.

**Acknowledgments**

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

# A    Gradient Calculation for LossVal

We discuss the difference between the gradients resulting from the variant of LossVal where target loss and distribution distance are summed up, i.e. LossVal$^+$, and the variant where they are multiplied, i.e. LossVal$^\bullet$. The gradients of the loss with respect to the instance-specific weights are the basis for updating the weights during training. Obtaining a better understanding of how the weights are computed improves our intuition about why using the multiplication in LossVal$^\bullet$ works better. We also leave out the squaring of the distribution distance $\text{OT}$ $\text{OT}_w$ for simplicity of the derivatives.

Computing the gradient for the additive variant

$$\text{LossVal}^+ = \mathcal{L}_w(y, \hat{y}) + \text{OT}_w(X_{\text{train}}, X_{\text{test}})$$

with respect to the weight $w_i$ of an instance $i$ is simply

$$\frac{\partial \text{LossVal}^+}{\partial w_i} = \frac{\partial \mathcal{L}_w}{\partial w_i} + \frac{\partial \text{OT}_w}{\partial w_i} \,. \tag{7}$$

For the multiplicative variant

$$\text{LossVal}^\bullet = \mathcal{L}_w(y, \hat{y}) \cdot \text{OT}_w(X_{\text{train}}, X_{\text{test}})$$

the chain rule is to be applied and results in

$$\frac{\partial \text{LossVal}^\bullet}{\partial w_i} = \left( \frac{\partial \mathcal{L}_w}{\partial w_i} \cdot \text{OT}_w \right) + \left( \mathcal{L}_w \cdot \frac{\partial \text{OT}_w}{\partial w_i} \right) \,. \tag{8}$$

It is easy to see how the weights $w_j$ for instances $j$ with $j \neq i$ get dropped in the gradient in Equation (7), but it persists in the gradient in Equation (8).

We briefly have a closer look at $\frac{\partial \mathcal{L}_w}{\partial w_i}$ and $\frac{\partial \text{OT}_w}{\partial w_i}$ for the case of using the MSE loss. $N$ is the number of training samples, and $J$ is the number of validation samples. We obtain

$$\frac{\partial \mathcal{L}_w}{\partial w_i} = \frac{\partial}{\partial w_i} \sum_{n=1}^{N} w_n \cdot (y_n - \hat{y}_n)^2 = (y_i - \hat{y}_i)^2$$

and for OT with the cost function $c(x_n, x_j)$, we obtain

$$\frac{\partial \text{OT}_w}{\partial w_i} = \frac{\partial}{\partial w_i} \sum_{n=1}^{N} \sum_{j=1}^{J} w_n \cdot c(x_n, x_j) = \sum_{j=1}^{J} c(x_i, x_j) \,.$$

Note that we assume that we already found the optimal transportation plan $\gamma*$.

Going back to Equation (7) for LossVal$^+$, we find that

$$\frac{\partial \mathcal{L}_w}{\partial w_i} + \frac{\partial \text{OT}_w}{\partial w_i} = (y_i - \hat{y}_i)^2 + \sum_{j=1}^{J} c(x_i, x_j) \,. \tag{9}$$

For Equation (8), we find for LossVal$^\bullet$ that

$$\left( \frac{\partial \mathcal{L}_w}{\partial w_i} \cdot \text{OT}_w \right) + \left( \mathcal{L}_w \cdot \frac{\partial \text{OT}_w}{\partial w_i} \right)$$

$$= (y_i - \hat{y}_i)^2 \cdot \text{OT}_w + \mathcal{L}_w \cdot \sum_{j=1}^{J} c(x_i, x_j) \,. \tag{10}$$

We observe that for the additive variant LossVal$^+$ in Equation (9), the gradient (and therefore the weight updates during training) only depends on the *local* loss for datapoint $i$ and the *local* optimal transport distance for the datapoint $i$. However, for the multiplicative variant of LossVal$^\bullet$ in Equation (10), the gradient depends not only on the local loss and local distance but on the overall loss and the overall optimal transport distance. Here, all instance-specific weights take part in updating each individual weight $w_i$, potentially making the gradient more informative.

## B    Theoretical Intuition

This section provides a simplified formal perspective on why LossVal assigns higher weights to "high-quality" samples and how the optimal transport (OT) term regularizes weights toward the validation feature distribution. The analysis is intended to be illustrative and explanatory rather than a formal equivalence result.

**Setup.**   Consider linear regression with heteroscedastic noise

$$y_i = \theta^\top x_i + \varepsilon_i, \qquad \varepsilon_i \sim \mathcal{N}(0, \sigma_i^2), \qquad i = 1, \ldots, N. \tag{11}$$

Let $X \in \mathbb{R}^{N \times d}$ be the design matrix (rows $x_i^\top$) and $y \in \mathbb{R}^N$.

**Weighted least squares intuition (no distribution shift).**   Assume there is no distribution shift between training and validation, in the sense that the feature distributions coincide or differ only negligibly. In this regime, the OT term used by LossVal is approximately constant with respect to the weights and therefore does not affect their optimum. The dominant effect of LossVal then comes from learning instance weights for a weighted least squares objective.

**Classical result (BLUE).**   If the noise covariance is $\mathrm{Cov}(\varepsilon) = \Sigma = \mathrm{diag}(\sigma_1^2, \ldots, \sigma_N^2)$ and $X$ has full column rank, the generalized least squares (GLS) estimator

$$\hat{\theta}_{\mathrm{GLS}} = \arg\min_\theta (y - X\theta)^\top \Sigma^{-1} (y - X\theta) = (X^\top \Sigma^{-1} X)^{-1} X^\top \Sigma^{-1} y \tag{12}$$

is the best linear unbiased estimator (BLUE). Equivalently, assigning sample weights proportional to the inverse noise variance, $\tilde{w}_i \propto 1/\sigma_i^2$, yields the optimal linear estimator under heteroscedastic Gaussian noise.

**Connection to LossVal weight dynamics.**   LossVal does not explicitly estimate noise variances, nor does it solve a generalized least squares or iteratively reweighted least squares problem.[1] However, its weight dynamics exhibit a closely related behavior. In the linear regression setting with mean squared error, the gradient of the weighted loss with respect to a sample weight $w_i$ is proportional to the squared residual $(y_i - \theta^\top x_i)^2$. Consequently, gradient-based weight updates tend to decrease the weights of samples with large residuals and increase the weights of samples with small residuals.

Under the no-shift assumption, this mechanism mirrors the classical inverse-variance weighting intuition behind BLUE: samples that appear noisier (large residuals) are down-weighted, while samples that are well explained by the model receive higher weight. This correspondence is heuristic rather than exact, but it explains why LossVal assigns higher importance to clean, informative samples even in the absence of feature distribution shift.

**OT regularization toward the validation feature distribution.**   When a distribution shift between training and validation features is present, LossVal incorporates an optimal transport term that depends on the learned weights. Let

$$\mu_{\mathrm{train}}^w := \sum_{i=1}^N w_i \, \delta_{x_i}, \qquad \mu_{\mathrm{val}} := \frac{1}{J} \sum_{j=1}^J \delta_{\tilde{x}_j} \tag{13}$$

---

[1]LossVal optimizes instance weights via gradient descent on a multiplicative objective that couples the weighted target loss and a weighted optimal transport term under simplex constraints. While the resulting weight dynamics resemble inverse-variance or IRLS-style behavior in simple settings, LossVal is neither likelihood-based nor equivalent to GLS or IRLS.

denote the weighted empirical training feature distribution and the empirical validation feature distribution, respectively. The OT component can be written as

$$\text{OT}(\mu_{\text{train}}^w, \mu_{\text{val}}) = \min_{\gamma \in \Pi(\mu_{\text{train}}^w, \mu_{\text{val}})} \sum_{i=1}^{N} \sum_{j=1}^{J} \gamma_{ij}\, c(x_i, \tilde{x}_j), \tag{14}$$

where $\Pi(\cdot, \cdot)$ denotes the set of couplings with prescribed marginals.

In LossVal, this OT term is multiplied with the weighted target loss rather than added. As a result, weight updates are influenced jointly by prediction error and feature-space alignment. Placing large weight on training samples whose features are far from the validation distribution increases the transport cost and is therefore penalized during optimization.

**Effect on gradients.** The gradient expressions derived in Appendix A make this interaction explicit. Each weight update depends on both the local prediction error and the global transport cost induced by the optimal coupling. Because the OT term couples all weights through the transport plan, the resulting updates are global rather than independent: reweighting one sample affects the feature alignment of the entire weighted training distribution. This explains how LossVal simultaneously down-weights noisy samples and shifts mass toward training data that better matches the validation feature distribution.

## C   Vehicle Crash Tests Background

In this section, we introduce the passive car safety scenario to demonstrate the benefit of data valuation for active data acquisition. Improving the crashworthiness and the restraint systems of a car is fundamental for saving the lives of the occupants and reducing injuries in a collision. To develop optimal passive restraint systems, such as airbags and seatbelts, engineers traditionally rely on physical crash tests or virtual simulations (Anonymized 1, 2024). However, these tests and simulations, while invaluable, are prohibitively expensive. A single crash test costs hundreds of thousands of dollars, and high-fidelity simulations cost hundreds of dollars each (Spethmann et al., 2009). The high costs and execution time associated with crash tests and simulations limit the number of them.

To mitigate these challenges, recent advancements have turned to machine learning models as surrogate tools for crash testing (Anonymized 3, 2021; Liu et al., 2023; Mathieu et al., 2024; Anonymized 2, 2022; Sun et al., 2023; Anonymized 1, 2024). By training models to predict the crash severity based on vehicle parameters, engineers can virtually assess and optimize safety features. Using machine learning models as surrogates for crash tests and simulations allows them to try out more different restraint system configurations and find good solutions faster. However, the effectiveness of these surrogate models is highly dependent on the quality and relevance of the training data used (Budach et al., 2022; Chen et al., 2021; Anonymized 1, 2024).

Publicly available crash test data goes back 40 years (National Highway Traffic Safety Administration (NHTSA), 2024; European New Car Assessment Programme (Euro NCAP), 2024), and progress in car design, materials, and technologies means that older results are not necessarily transferable to modern cars. To improve the machine learning models for current cars and prototypes, we need to identify which crash tests are beneficial and determine if additional training data, like crash tests and simulations, is needed. Understanding the importance of an individual data point for the model's performance is crucial for prioritizing the acquisition of new data that offers the greatest potential for enhancing predictive accuracy.

## D   Details of the Datasets

### D.1   Classification Datasets

Table 5 presents an overview of the six datasets for classification tasks. Those tabular datasets are widely used in the literature and are the focus of the OpenDataVal benchmark (Jiang et al., 2023). Each dataset is standardized before use.

Table 5: The classification datasets we used. *fried* and *2dplanes* are binarized.

| Dataset | Sample Size | Input Dimension | Number of Classes | Minor Class Proportion | Source |
|---|---|---|---|---|---|
| electricity | 38,474 | 6 | 2 | 0.50 | (Gama et al., 2004) |
| fried | 40,768 | 10 | 2 | 0.50 | (Friedman, 1991) |
| 2dplanes | 40,768 | 10 | 2 | 0.50 | (Breiman et al., 2017) |
| pol | 15,000 | 48 | 2 | 0.37 | OpenML-722 |
| MiniBooNE | 72,998 | 50 | 2 | 0.50 | (Roe et al., 2005) |
| nomao | 34,365 | 89 | 2 | 0.29 | (Candillier & Lemaire, 2012) |

## D.2 Regression Datasets

The OpenDataVal benchmark does not include predefined regression datasets, so we selected six datasets from the CTR-23 benchmark suite (Fischer, 2023) according to specific criteria. We ensured that all selected datasets contain only numeric features, have no missing values, and include at least 4,100 samples ($1,000$ for training, 100 for validation, and $3,000$ for testing). Additionally, for datasets with fewer than 45 features, we limited the maximum number of samples to $10,000$. The resulting six regression datasets are similar in numbers of features and samples to the classification datasets, as described in Table 6. Like the classification datasets, these were standardized before use.

Table 6: Description of a subset of the regression datasets from the CTR23 benchmark suite we used Fischer (2023).

| Dataset | Sample Size | Input Dimension | Mean | Standard Deviation | OpenML ID |
|---|---|---|---|---|---|
| kin8nm | 8,192 | 8 | 0.71 | 0.26 | 44980 |
| white_wine | 4,898 | 11 | 5.88 | 0.89 | 44971 |
| cpu_activity | 8,192 | 21 | 83.97 | 18.40 | 44978 |
| pumadyn32nh | 8,192 | 32 | 0 | 0.04 | 44981 |
| wave_energy | 72,000 | 48 | 3,760,135 | 112,145 | 44975 |
| superconductivity | 21,263 | 81 | 34.42 | 34.25 | 44964 |

## D.3 Crash Test Dataset

We use a dataset with vehicle crash tests to evaluate the effectiveness of LossVal in active data acquisition. This dataset, created to support the development of restraint systems for vehicles, consists of $1,122$ publicly available crash tests from the National Highway Traffic Safety Administration (NHTSA) (National Highway Traffic Safety Administration (NHTSA), 2024) and 154 proprietary crash tests provided by a large car manufacturer (Anonymized 3, 2021; Anonymized 2, 2022). For this study, we focus on full-frontal crash tests conducted at 56 km/h (about 15.6 m/s). The data contains numerous metrics and sensor data, from which we extract the features described in Section D.4. All features derive from vehicle information or sensors built into the car.

Our goal is to predict the injury severity for the occupant (in our case, the dummy) from car-bound information alone (without any information from the dummy). The target variable is the Real Occupant Load Criterion ($ROLC_p$), an adapted variant of the Occupant Load Criterion (OLC) (Anonymized 2, 2022; Anonymized 1, 2024). The $ROLC_p$ is calculated from the dummy chest acceleration and highly correlated with the load on the dummy during the crash test. Our goal is to predict the $ROLC_p$ using only car-specific features without knowing the acceleration signals of the dummy.

According to the $ROLC_p$-Model, a vehicle crash can be divided into three phases, as shown in Figure 5 in Section D.4. After impact, the car decelerates (between 0 and $t_1$ on the time axis), but the dummy does not decelerate immediately because the dummy and car are not rigidly connected. There is some space between the belt and the dummy chest. As the car decelerates, the dummy continues moving at the original speed until the dummy is connected to the vehicle deceleration via the restraint system. The moment of coupling is called $t_1$. The dummy speed at $t_1$ is equal to $v_1$. Between $t_1$ and $t_2$, the dummy experiences a deceleration.

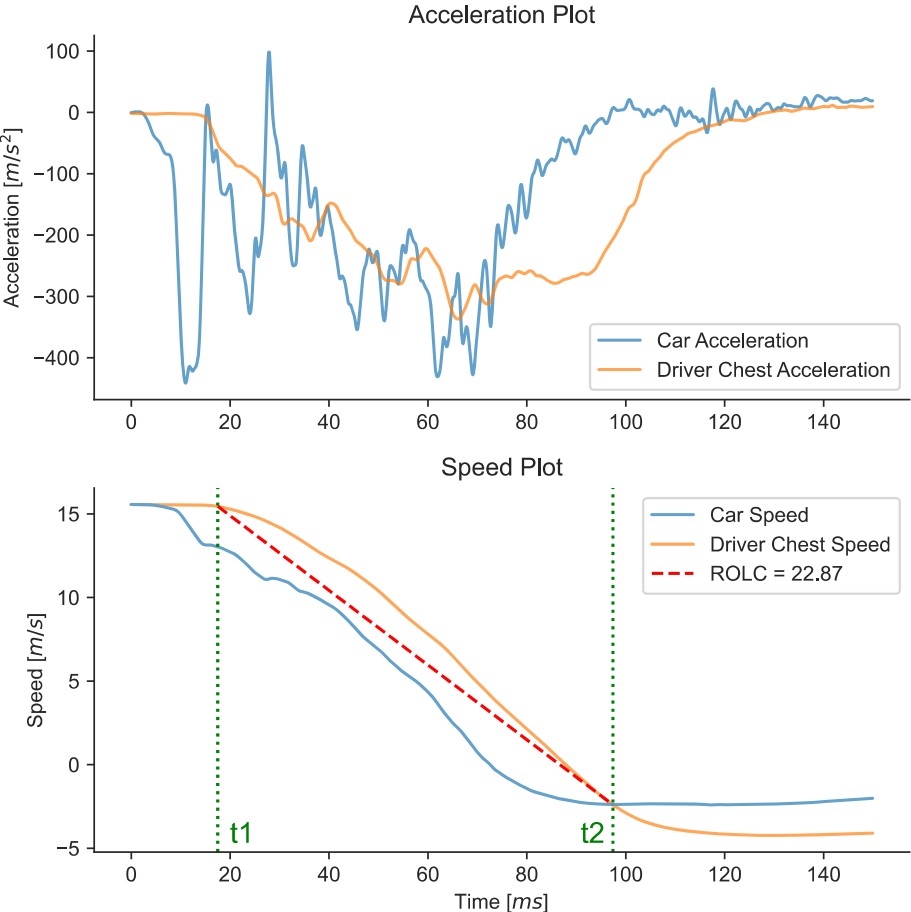

Figure 5: Exemplary crash test showing vehicle and occupant acceleration (top) and speed (bottom), as well as the corresponding $ROLC_p$ model.

$t_2$ is defined as the moment when driver and car speed are equal with a constant rebound speed ($v_2$). The $ROLC_p$ is defined as the absolute slope of a line from point $A(t_1, v_1)$ to point $B(t_2, v_2)$, measured in $g$. We refer to (Anonymized 2, 2022; Anonymized 1, 2024) for a more extensive discussion of the $ROLC_p$.

### D.4 Example, Features, and Configurable Parameters of a Crash Test

**Example** Figure 5 shows the sensor signals from an exemplary crash test.

**Features** The features of a crash test are in detail:

**Car acceleration.** The car acceleration signal over 130 ms, sampled every 2 ms.

**Car body type.** One-hot encoding of the car body type (selection from: convertible, pickup truck, four-door sedan, van, five-door hatchback, utility vehicle, three-door hatchback, two-door coupe, two-door sedan, extended cap pickup, minivan, four-door pickup, station wagon, truck, three-door coupe).

**Car mass.** Mass of the vehicle in kg.

**Car speed at $t0$.** The speed of the car at the moment of impact. It slightly varies, but is always around 15.6 m/s.

**Restraint system time-to-fire.** How many milliseconds after impact, the restraint system components (airbag, belt tensioner) fire.

**Chest to steering wheel distance.** The distance between steering wheel and driver chest.

**Number of shoulder belt force limiters.** Either 0, 1, or 2.

**Shoulder belt force level 1.** Threshold of the first level belt force limiter.

**Shoulder belt force level 2.** Threshold of the second level belt force limiter.

**Shoulder belt force limiter switching time.** The point in time when switching from the first to the second belt force limiter.

**Availability of the shoulder belt pretensioner.** Either 1 when a pretensioner is available or 0, otherwise.

**Average car acceleration.** Average car acceleration considering only the x-axis (the driving direction).

**Maximum car acceleration.** Maximum car acceleration considering only the x-axis.

**Maximum car acceleration over 3ms.** Maximum car acceleration considering only the x-axis and only accelerations endured for longer than 3 ms (flattening high peaks).

**Dynamic deformation.** Maximum dynamic deformation (Huang et al., 1995).

**Kinetic energy.** The kinetic energy of the car on impact.

**$SM_{25ms}$.** Sliding mean over 25 ms (Gu et al., 2005).

**TTZV.** Time to zero velocity (Viano & Arepally, 1990).

**OLC.** Occupant Load Criterion (Kübler et al., 2009).

**OLC++.** Linear combination of OLC, $SM_{25ms}$ and TTZV (Kübler et al., 2009).

**MCD.** Mean crash deceleration Anonymized 1 (2024).

**$\Delta V$.** Maximum velocity difference (Wu et al., 2002).

**Rebound velocity.** Maximum rebound speed after impact.

**Configurable Parameters** In the following, we give the features used for training the secondary model in the active data acquisition experiments. We limit the features used by the secondary model because we want to simulate a guided data acquisition process, where new crash tests are executed. Of course, before a new crash test is executed, we do not know the occupant load. This includes, for example, the weight of the car and the belt force limiter, so we can't use them for predicting the expected value of the crash test. We use only the following features when training the secondary model:

- Car body type.
- Car mass.
- Restraint system time-to-fire.
- Chest to steering wheel distance.
- Number of shoulder belt force limiters.
- Shoulder belt force level 1.
- Shoulder belt force level 2.
- Availability of the shoulder belt pretensioner.

# E  Detailed Procedure of the Active Data Acquisition Task

We provide details about the active data acquisition experiment using the crash test dataset. The process is shown in Figure 6. Since it is not feasible to generate new crash tests for this study, we simulate the data acquisition process using the existing dataset, by taking the highest-expected value data point from an unseen acquisition set.

First, we train a crash model (the MLP optimized for the crash test dataset) on the training data to predict the $ROLC_p$. Then, we use LossVal and the baseline data valuation methods to estimate importance scores

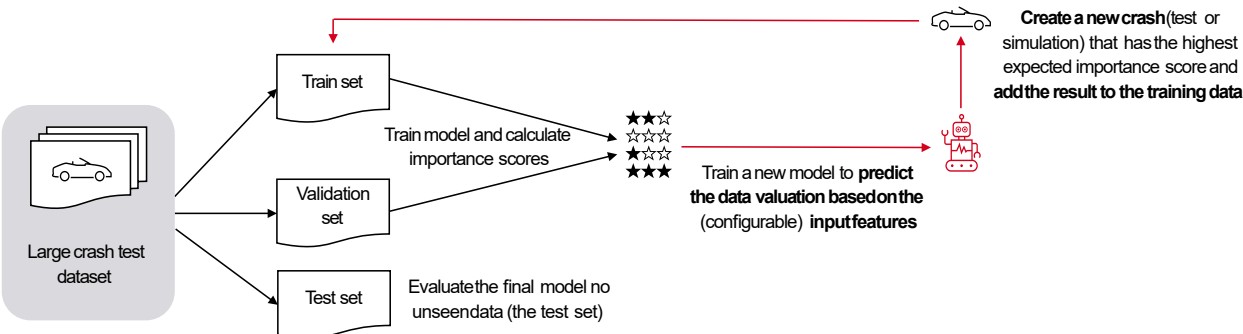

Figure 6: Experimental setup for the active data acquisition.

for all training samples. For each method, we train a secondary model (random forest) to predict the importance score based on the features of the corresponding training sample. The secondary model cannot "see" all the features of the data, but only the *configurable* features. This means the features that are known or can be changed before a crash test is executed, including, for example, the weight of the car and the belt force limiter (the full list is given in Section D.4). This procedure allows us to simulate an active learning approach that prioritizes data points based on their contribution to improving the model's performance.

The secondary model is used to predict the expected importance scores of the data points in the acquisition set. We take the 1% data points with the highest expected importance scores from the acquisition set and add them to the training set. Then we train the crash model again with the extended training set and compare the test performance of the model before and after adding more training data. The training of the crash model is repeated 10 times each to reduce the effects of randomness, but the random forest is only fitted once. The whole procedure is repeated 15 times.

## F    Hyperparameters

### F.1    Hyperparameter Search Space for the MLP

**Number of hidden layers:** $1, 2, 3, 4, 5$.

**Size of hidden layers:** $10, 20, \ldots, 100$.

**Learning rate:** $0.001, 0.01, 0.1$.

**Batch size:** $32, 64, 128$.

**Activation function:** tanh, sigmoid, ReLU

### F.2    Overview of other Hyperparameters

We collect all hyperparameters used in the experiments here for reference.

**OpenDataVal Benchmark Experiments:**

**Training / Validation / Test split:** 1000 / 100 / 3000 samples.

**Noise rates:** 5%, 10%, 15%, 20%.

**Gaussian noise parameters:** $\mu = 0, \sigma = 1.0$.

**Number of models for AME:** 1000.

**Number of models for Data-OOB:** 1000.

**Number of training epochs for DVRL:** 2000.

**Number of neighbors $k$ for KNN-Shapley:** 100.

**Number of experiment repetitions:** ~~25.~~ 15.

**Number of training epochs:** 5 (exception: LossVal is trained for 5 and for 30 epochs).

**MLP hyperparameters:** See Table 7.

**Active Data Acquisition Experiment:**

**Dataset size:** 1276 samples.

**Training / Validation / Acquisition / Test split:** 40% / 10% / 40% / 10% of the dataset.

**Newly acquired samples added to training set:** 1% of the acquisition set.

**Number of experiment repetitions:** ~~50.~~ 15.

**MLP training repetitions:** 10.

**Secondary model:** Random forest regressor with 100 estimators.

**Number of models for AME:** Equal to the training set size.

**Number of models for Data-OOB:** Equal to the training set size.

**Number of training epochs for DVRL:** Equal to 2 times the training set size.

**Number of neighbors $k$ for KNN-Shapley:** 10% of the training set size.

**Number of training epochs:** 5 (exception: LossVal is trained for 5 and for 30 epochs).

**MLP hyperparameters:** See Table 7.

### F.3 Optimal MLP Hyperparameter Values

The optimal MLP hyperparameter configurations are given in Table 7 and used in all experiments.

Table 7: The MLP hyperparameters we found to achieve the best performance for the three different tasks.

|  | Classification Benchmark | Regression Benchmark | Crash Scenario |
|---|---|---|---|
| Size of hidden layers | 100 | 90 | 100 |
| Number of hidden layers | 5 | 3 | 3 |
| Activation function | ReLU | tanh | tanh |
| Learning rate | 0.1 | 0.01 | 0.01 |
| Batch size | 128 | 32 | 32 |

### F.4 Optimal Transport Hyperparameters

Finetuning the hyperparameters for the Optimal Transport term in the loss function had a negligible effect on the task performance of LossVal. To avoid increasing the complexity of the setup for marginal gains, we chose reasonable default values for the hyperparameters of the Sinkhorn loss and used them for all the experiments. The Sinkhorn implementation of the `geomloss` Python package was used.

**Cost function:** Squared euclidean distance ($\Rightarrow p = 2$).

**Scaling:** 0.5

**Entropic Regularization:** $\varepsilon = 0.1$

**Blur:** $\varepsilon^{1/p}$

**Diameter:** $1.1\times$ the maximum pair-wise distance between points in train and validation set.

# G Extended Results

## G.1 Noisy Sample Detection Curves

Figure 9 shows the noisy sample detection curves from the noisy sample detection experiment. The curve shows the proportion of noisy samples detected per proportion of data inspected, with the underlying assumption that noisy data points will receive the lowest importance scores. Say, we add noise to 20% of the data points. Then a perfect data valuation method would therefore have detected 25% of the noisy data points, after inspecting 5% of all data points (starting with the data points with the lowest importance score). Table 9 gives the average over each curve, dataset, and noise rate.

Table 11 and Table 12 show the noisy label detection F1 scores from Section 5 broken down by dataset. They largely reflect the results from Table 2.

## G.2 Average of the Point Addition and Removal Curves

For better comparability, we give the averages of all the curves in Table 10. For regression, the negative MSE was normalized by dividing all values by the maximum value. This makes the table more readable because the experiment resulted in very large negative MSE values. Lower values are better for both high value point removal and low value point addition, because this indicates a faster decrease in test performance when data points with a high importance score are removed from the training set (or a slower increase in performance, when bad data points are added to the training set, respectively).

## G.3 Active Data Acquisition

Table 8 shows the results of the Active Data Acquisition results.

Table 8: Comparison of the test performance for active data acquisition. The "baseline" shows the test performance before adding new data, "random" reflects the effect of randomly adding data.

|  | MSE ($\pm$ SE) | MAPE ($\pm$ SE) | $R^2$ ($\pm$ SE) |
|---|---|---|---|
| Baseline | 0.234$\pm$0.008 | **0.892**$\pm$0.047 | 0.168$\pm$0.034 |
| Random | 0.237$\pm$0.007 | 0.935$\pm$0.048 | 0.160$\pm$0.027 |
| AME | **0.231**$\pm$0.009 | 0.899$\pm$0.046 | **0.182**$\pm$0.033 |
| Beta Shapley | 0.232$\pm$0.008 | 0.925$\pm$0.045 | *0.175*$\pm$0.034 |
| DVRL | 0.242$\pm$0.010 | 0.929$\pm$0.042 | 0.142$\pm$0.042 |
| Data Banzhaf | 0.246$\pm$0.010 | 0.960$\pm$0.047 | 0.122$\pm$0.044 |
| Data-OOB | 0.233$\pm$0.010 | 0.930$\pm$0.049 | 0.175$\pm$0.036 |
| Data Shapley | *0.233*$\pm$0.009 | 0.935$\pm$0.043 | 0.171$\pm$0.036 |
| Influence Subsample | 0.237$\pm$0.008 | 0.937$\pm$0.046 | 0.159$\pm$0.030 |
| KNN-Shapley | 0.240$\pm$0.009 | 0.930$\pm$0.049 | 0.149$\pm$0.037 |
| LAVA | 0.246$\pm$0.009 | 0.931$\pm$0.043 | 0.122$\pm$0.045 |
| Leave-One-Out | 0.243$\pm$0.010 | 0.945$\pm$0.045 | 0.137$\pm$0.042 |
| LossVal (epochs = 5) | 0.242$\pm$0.008 | *0.912*$\pm$0.042 | 0.137$\pm$0.038 |
| LossVal (epochs = 30) | 0.244$\pm$0.009 | 0.938$\pm$0.048 | 0.131$\pm$0.039 |

## G.4 Importance Score Distribution

Figure 7 shows how the importance scores are distributed for each method. KNN-Shapley did fail to find useful importance scores. Figure 8 shows the normalized value of the importance scores, when sorted by the value.

## G.5 Data-OOB Comparison

After finishing our experiments, it seemed like Data-OOB (Kwon & Zou, 2023) performed worse in our experiments than in the results provided by Jiang et al. (2023). Upon investigation, we found that using an MLP instead of logistic regression as the base model for classification tasks leads to a decreased performance

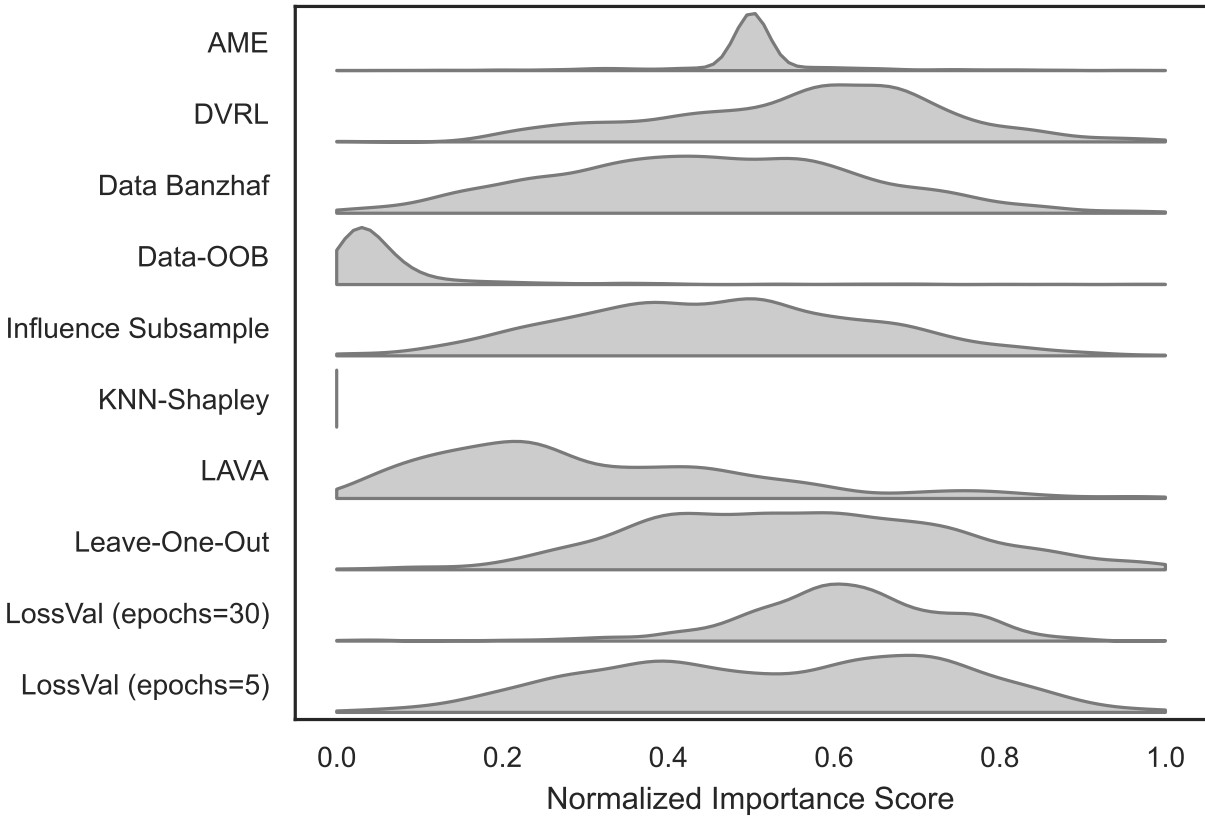

Figure 7: This plot shows the density of the normalized importance scores of each method.

in the data valuation. We repeated the experiments for Data-OOB using logistic regression and linear regression as base models for the noisy sample detection. Figure 10 and Figure 11 show that using logistic regression works much better for Data-OOB than using an MLP or linear regression. Still, LossVal achieves similar or better results compared to Data-OOB for both regression and classification.

## H Runtime Analysis

Table 13 shows a runtime comparison between the baselines used in this study and LossVal. The measurement was repeated five times on a RTX 3060 GPU. Note that LAVA and KNN-Shapley were executed on an 8-core CPU instead of the GPU, because they are model-free and do not train an MLP as the other methods do. Accordingly, observe that they are faster than the other methods, because running them needs less time than training a single MLP. Aside from those two, LossVal with 5 and LossVal with 30 epochs are significantly faster than the baselines.

We observe that Data Shapley is faster than Data-OOB, because the implementation of Data Shapley uses an approximation instead of calculating the exact Shapley values. The table also shows that the real-world performance of Data-OOB, Leave-One-Out, and Influence Subsample differ, even though have the same number of training runs. This stems from the fact, that they use subsets of different size for retraining, affecting the duration of each training run.

The runtime comparison for the large scale evaluation with BERT and ResNet50 is given in Table 14. They were executed on a V100 GPU and a 20-core CPU. KNN-Shapely and LAVA are very fast, because they do not need to train a model. However, for the text-based 20Newsgroups datasets, they work on the embeddings of the text samples. The training of the base model that is used to create the embeddings is not included in the training. The complexity of LossVal depends upon the number of training epochs and the complexity of

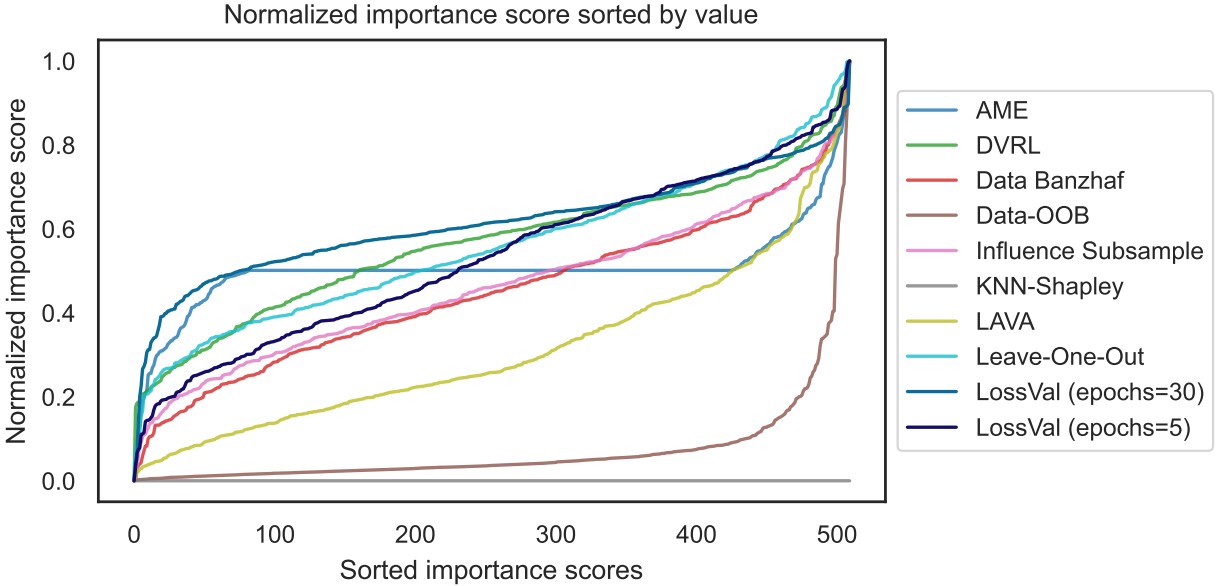

Figure 8: This plot shows the normalized importance scores sorted for each method. The y-axis is the value of the importance score. They are sorted along the x-axis.

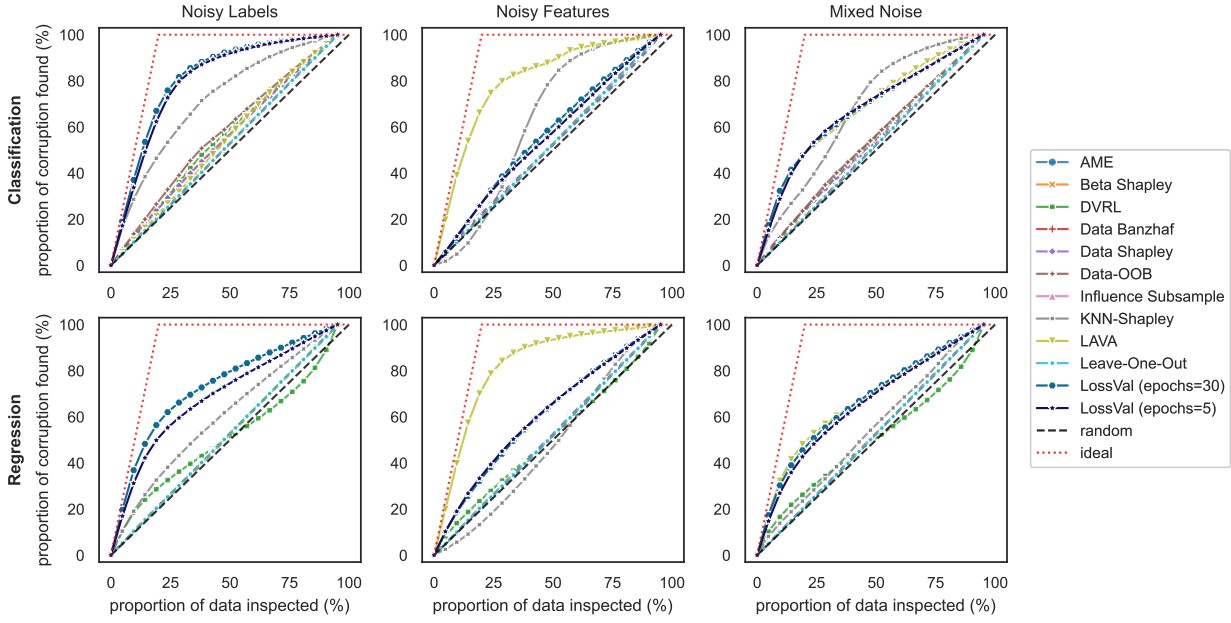

Figure 9: Noisy sample detection for classification (top row) and regression (bottom row). The curves show the average for all classification and regression datasets, respectively. Higher is better. In the lower plot, certain methods are obscured along the random line.

|  |  | Mixed Noise (%) | Noisy Features (%) | Noisy Labels (%) | Overall Average (%) |
|---|---|---|---|---|---|
| **Classification** | AME | 49.98±0.27 | 49.81±0.27 | 50.00±0.27 | 49.93±0.16 |
|  | Beta Shapley | 53.32±0.27 | 50.13±0.26 | 51.76±0.27 | 51.74±0.16 |
|  | DVRL | 53.87±0.28 | 49.97±0.27 | 52.07±0.27 | 51.97±0.16 |
|  | Data Banzhaf | 50.35±0.27 | 49.97±0.27 | 50.04±0.27 | 50.12±0.16 |
|  | Data-OOB | 56.52±0.28 | 50.25±0.27 | 53.19±0.27 | 53.32±0.16 |
|  | Data Shapley | 53.25±0.27 | 50.12±0.26 | 51.69±0.27 | 51.69±0.16 |
|  | Influence Subsample | 50.34±0.27 | 49.96±0.27 | 50.09±0.27 | 50.13±0.16 |
|  | KNN-Shapley | _72.49_±0.26 | 62.78±0.35 | 68.22±0.29 | _67.83_±0.18 |
|  | LAVA | 54.75±0.29 | **81.18**±0.24 | **68.76**±0.24 | 68.23±0.16 |
|  | Leave-One-Out | 50.12±0.27 | 49.85±0.27 | 49.96±0.27 | 49.98±0.16 |
|  | LossVal (epochs=5) | 81.61±0.25 | 53.96±0.27 | 67.29±0.23 | 67.62±0.16 |
|  | LossVal (epochs=30) | **83.10**±0.25 | _55.13_±0.28 | _67.91_±0.23 | **68.72**±0.16 |
| **Regression** | AME | 49.85±0.27 | 50.04±0.27 | 49.87±0.27 | 49.92±0.16 |
|  | Beta Shapley | 50.14±0.27 | 50.10±0.27 | 50.15±0.27 | 50.13±0.16 |
|  | DVRL | 49.66±0.29 | 50.00±0.27 | 49.25±0.27 | 49.63±0.16 |
|  | Data Banzhaf | 49.85±0.27 | 50.04±0.27 | 49.87±0.27 | 49.92±0.16 |
|  | Data-OOB | 49.85±0.27 | 50.04±0.27 | 49.87±0.27 | 49.92±0.16 |
|  | Data Shapley | 50.14±0.27 | 50.10±0.27 | 50.15±0.27 | 50.13±0.16 |
|  | Influence Subsample | 49.85±0.27 | 50.04±0.27 | 49.87±0.27 | 49.92±0.16 |
|  | KNN-Shapley | _58.88_±0.28 | 47.33±0.33 | 53.11±0.29 | 53.10±0.17 |
|  | LAVA | 50.03±0.27 | **83.40**±0.24 | 66.90±0.23 | 66.77±0.16 |
|  | Leave-One-Out | 49.85±0.27 | 50.04±0.27 | 49.87±0.27 | 49.92±0.16 |
|  | LossVal (epochs=5) | 70.39±0.25 | 61.47±0.27 | 66.13±0.25 | _66.00_±0.15 |
|  | LossVal (epochs=30) | **74.23**±0.24 | _61.18_±0.28 | **67.80**±0.25 | **67.74**±0.15 |

Table 9: Average of the corruption discovery curves of each data valuation method, averaged over all proportion steps, noise rates, and datasets. The number after ± indicates the standard error. Higher is better.

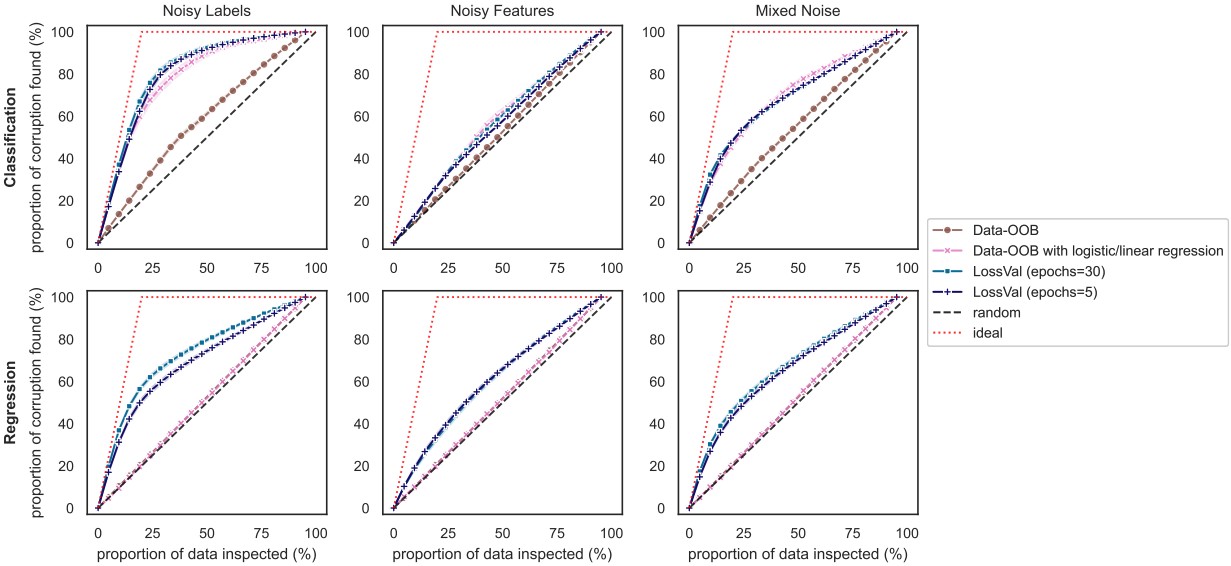

Figure 10: Noisy sample detection for classification (top row) and regression (bottom row). The curves show the average for all classification and regression datasets, respectively. Higher is better.

Table 10: Results of the point removal and addition experiment (compare Figure 3 and Figure 4), averaged over all removal rates from $0 - 50\%$ (left) and addition rates from $5 - 50\%$ (right) and all classification datasets (top) and regression datasets (bottom). The number after $\pm$ indicates the standard error. Lower is better.

| | **Point Removal Experiment** | | | | **Point Addition Experiment** | | | |
| | Noisy Labels | Noisy Features | Mixed Noise | Overall Average | Noisy Labels | Noisy Features | Mixed Noise | Overall Average |
|---|---|---|---|---|---|---|---|---|
| **Classification** | | | | | | | | |
| AME | 74.86±0.10 | 75.31±0.10 | 75.08±0.10 | 75.08±0.06 | 74.86±0.10 | 75.31±0.10 | 75.08±0.10 | 75.08±0.06 |
| Beta Shapley | 73.06±0.11 | 74.40±0.10 | 73.91±0.11 | 73.79±0.06 | 73.06±0.11 | 74.40±0.10 | 73.91±0.11 | 73.79±0.06 |
| DVRL | 70.41±0.12 | 72.37±0.11 | *71.74*±0.11 | *71.50*±0.07 | 70.41±0.12 | 72.37±0.11 | *71.74*±0.11 | *71.50*±0.07 |
| Data Banzhaf | 74.66±0.10 | 75.25±0.10 | 74.95±0.10 | 74.96±0.06 | 74.66±0.10 | 75.25±0.10 | 74.95±0.10 | 74.96±0.06 |
| Data-OOB | 70.45±0.12 | *72.55*±0.11 | 72.02±0.11 | 71.67±0.07 | 70.45±0.12 | *72.55*±0.11 | 72.02±0.11 | 71.67±0.07 |
| Data Shapley | 73.32±0.11 | 74.48±0.10 | 73.99±0.10 | 73.93±0.06 | 73.32±0.11 | 74.48±0.10 | 73.99±0.10 | 73.93±0.06 |
| Influence Subsample | 74.74±0.10 | 75.29±0.10 | 75.01±0.10 | 75.02±0.06 | 74.74±0.10 | 75.29±0.10 | 75.01±0.10 | 75.02±0.06 |
| KNN-Shapley | **67.21**±0.14 | **71.38**±0.13 | **69.58**±0.13 | **69.39**±0.08 | **67.21**±0.14 | **71.38**±0.13 | **69.58**±0.13 | **69.39**±0.08 |
| LAVA | 71.62±0.11 | 73.78±0.11 | 73.28±0.11 | 72.89±0.06 | 71.62±0.11 | 73.78±0.11 | 73.28±0.11 | 72.89±0.06 |
| Leave-One-Out | 74.89±0.10 | 75.33±0.10 | 75.06±0.10 | 75.10±0.06 | 74.89±0.10 | 75.33±0.10 | 75.06±0.10 | 75.10±0.06 |
| LossVal (epochs=5) | 68.64±0.17 | 73.15±0.13 | 71.22±0.15 | 71.00±0.09 | 68.64±0.17 | 73.15±0.13 | 71.22±0.15 | 71.00±0.09 |
| LossVal (epochs=30) | *69.86*±0.17 | 73.57±0.12 | 72.01±0.14 | 71.81±0.08 | *69.86*±0.17 | 73.57±0.12 | 72.01±0.14 | 71.81±0.08 |
| **Regression** | | | | | | | | |
| AME | -0.143±0.001 | -0.128±0.001 | -0.129±0.001 | -0.133±0.001 | -0.143±0.001 | -0.128±0.001 | -0.129±0.001 | -0.133±0.001 |
| Beta Shapley | -0.146±0.001 | -0.122±0.001 | -0.134±0.001 | -0.134±0.001 | -0.146±0.001 | -0.122±0.001 | -0.134±0.001 | -0.134±0.001 |
| DVRL | **-0.173**±0.002 | **-0.151**±0.002 | **-0.154**±0.002 | **-0.160**±0.001 | **-0.173**±0.002 | **-0.151**±0.002 | **-0.154**±0.002 | **-0.160**±0.001 |
| Data Banzhaf | -0.143±0.001 | -0.128±0.001 | -0.129±0.001 | -0.133±0.001 | -0.143±0.001 | -0.128±0.001 | -0.129±0.001 | -0.133±0.001 |
| Data-OOB | -0.143±0.001 | -0.128±0.001 | -0.129±0.001 | -0.133±0.001 | -0.143±0.001 | -0.128±0.001 | -0.129±0.001 | -0.133±0.001 |
| Data Shapley | -0.146±0.001 | -0.122±0.001 | -0.134±0.001 | -0.134±0.001 | -0.146±0.001 | -0.122±0.001 | -0.134±0.001 | -0.134±0.001 |
| Influence Subsample | -0.143±0.001 | -0.128±0.001 | -0.129±0.001 | -0.133±0.001 | -0.143±0.001 | -0.128±0.001 | -0.129±0.001 | -0.133±0.001 |
| KNN-Shapley | *-0.163*±0.002 | *-0.137*±0.001 | -0.140±0.001 | *-0.147*±0.001 | *-0.163*±0.002 | *-0.137*±0.001 | -0.140±0.001 | *-0.147*±0.001 |
| LAVA | -0.148±0.002 | -0.148±0.001 | *-0.142*±0.001 | -0.146±0.001 | -0.148±0.002 | -0.148±0.001 | *-0.142*±0.001 | -0.146±0.001 |
| Leave-One-Out | -0.143±0.001 | -0.128±0.001 | -0.129±0.001 | -0.133±0.001 | -0.143±0.001 | -0.128±0.001 | -0.129±0.001 | -0.133±0.001 |
| LossVal (epochs=5) | -0.152±0.002 | -0.129±0.001 | -0.135±0.001 | -0.139±0.001 | -0.152±0.002 | -0.129±0.001 | -0.135±0.001 | -0.139±0.001 |
| LossVal (epochs=30) | -0.165±0.002 | -0.135±0.001 | -0.144±0.001 | -0.148±0.001 | -0.165±0.002 | -0.135±0.001 | -0.144±0.001 | -0.148±0.001 |

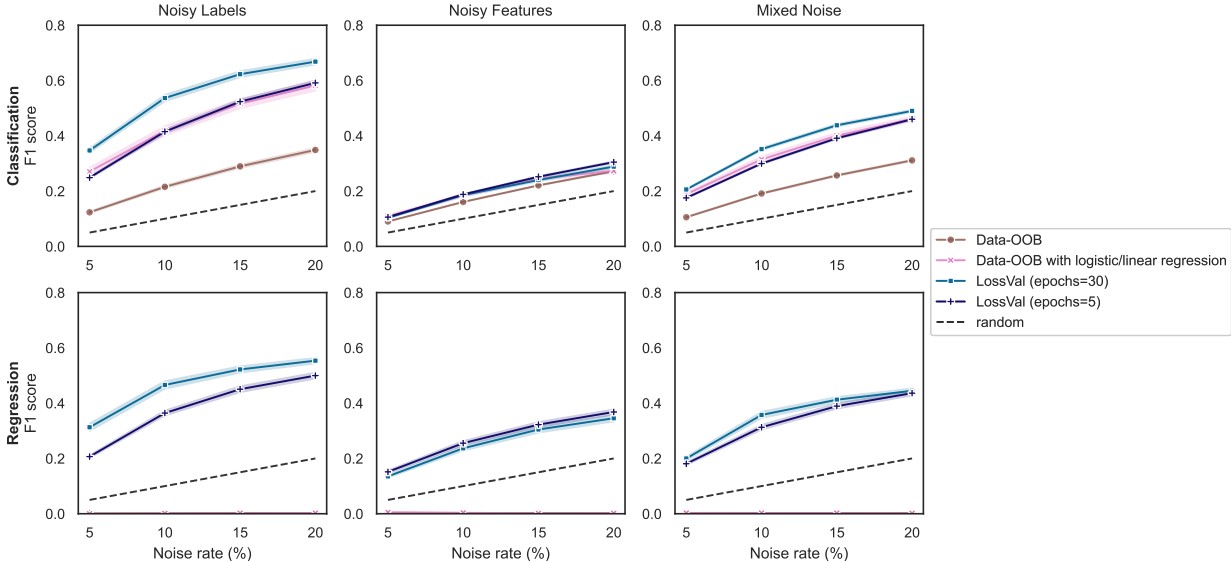

Figure 11: Noisy sample detection comparing Data-OOB with MLP and Data-OOB with logistic or linear regression. Higher is better.

DataInf depends on the number of parameters in the last layer and the number of training samples. We can observe this in the table: Both methods add more overhead to CIFAR-10 training than to the BERT training, because the ResNet on CIFAR-10 is trained for more epochs (relevant for LossVal) and the CIFAR-10 dataset contains more datapoints (relevant for DataInf), compared to BERT on 20Newsgroups.

Table 11: Average of the noisy sample detection F1 scores on the classification task, averaged over all noise rates. The number after ± indicates the standard error. Higher is better.

| | | Noisy Labels | Noisy Features | Mixed Noise | Overall Average |
|---|---|---|---|---|---|
| **2dplanes** | AME | 0.069±.012 | 0.091±.012 | 0.080±.011 | 0.080±.007 |
| | DVRL | 0.192±.008 | 0.191±.008 | 0.187±.008 | 0.190±.004 |
| | Data Banzhaf | 0.196±.007 | 0.194±.007 | 0.191±.008 | 0.193±.004 |
| | Data-OOB | 0.191±.007 | 0.193±.008 | 0.193±.007 | 0.192±.004 |
| | Influence Subsample | 0.193±.007 | 0.189±.008 | 0.189±.007 | 0.190±.004 |
| | KNN-Shapley | *0.344*±.013 | 0.274±.010 | *0.316*±.011 | 0.311±.007 |
| | LAVA | 0.213±.007 | **0.434**±.016 | 0.297±.011 | *0.315*±.009 |
| | Leave-One-Out | 0.183±.008 | 0.163±.010 | 0.179±.009 | 0.175±.005 |
| | LossVal (epochs=5) | 0.462±.014 | 0.236±.008 | 0.366±.012 | 0.356±.009 |
| | LossVal (epochs=30) | **0.592**±.014 | *0.243*±.008 | **0.427**±.012 | **0.423**±.011 |
| **electricity** | AME | 0.067±.011 | 0.090±.013 | 0.079±.012 | 0.079±.007 |
| | DVRL | 0.188±.008 | 0.186±.008 | 0.192±.008 | 0.189±.005 |
| | Data Banzhaf | 0.194±.007 | 0.193±.008 | 0.193±.008 | 0.193±.004 |
| | Data-OOB | 0.196±.007 | 0.196±.007 | 0.194±.007 | 0.195±.004 |
| | Influence Subsample | 0.193±.008 | 0.192±.008 | 0.191±.008 | 0.192±.004 |
| | KNN-Shapley | *0.289*±.010 | **0.216**±.008 | *0.255*±.009 | *0.253*±.006 |
| | LAVA | 0.023±.007 | 0.060±.014 | 0.034±.008 | 0.039±.006 |
| | Leave-One-Out | 0.180±.010 | 0.165±.009 | 0.176±.008 | 0.174±.005 |
| | LossVal (epochs=5) | 0.331±.011 | 0.207±.008 | 0.271±.010 | 0.270±.006 |
| | LossVal (epochs=30) | **0.359**±.012 | *0.201*±.008 | **0.282**±.010 | **0.281**±.007 |
| **fried** | AME | 0.077±.011 | 0.084±.012 | 0.091±.012 | 0.084±.007 |
| | DVRL | 0.190±.008 | 0.191±.008 | 0.190±.008 | 0.190±.004 |
| | Data Banzhaf | 0.189±.008 | 0.195±.008 | 0.190±.008 | 0.191±.004 |
| | Data-OOB | 0.190±.007 | 0.192±.007 | 0.196±.007 | 0.193±.004 |
| | Influence Subsample | 0.195±.007 | 0.194±.007 | 0.193±.007 | 0.194±.004 |
| | KNN-Shapley | *0.322*±.012 | 0.264±.010 | *0.295*±.010 | 0.294±.006 |
| | LAVA | 0.200±.007 | **0.419**±.016 | 0.284±.010 | *0.301*±.008 |
| | Leave-One-Out | 0.181±.009 | 0.181±.008 | 0.178±.009 | 0.180±.005 |
| | LossVal (epochs=5) | 0.437±.013 | 0.213±.008 | 0.331±.011 | 0.327±.008 |
| | LossVal (epochs=30) | **0.535**±.014 | *0.224*±.008 | **0.384**±.012 | **0.381**±.010 |
| **MiniBooNE** | AME | 0.093±.012 | 0.083±.011 | 0.088±.012 | 0.088±.007 |
| | DVRL | 0.185±.008 | 0.199±.011 | 0.190±.008 | 0.191±.005 |
| | Data Banzhaf | 0.194±.008 | 0.195±.008 | 0.194±.007 | 0.194±.004 |
| | Data-OOB | 0.209±.009 | 0.193±.008 | 0.194±.007 | 0.199±.005 |
| | Influence Subsample | 0.193±.008 | 0.192±.007 | 0.193±.007 | 0.193±.004 |
| | KNN-Shapley | 0.374±.013 | **0.368**±.012 | 0.365±.014 | **0.369**±.008 |
| | LAVA | 0.046±.010 | 0.094±.023 | 0.100±.018 | 0.080±.010 |
| | Leave-One-Out | 0.139±.010 | 0.171±.009 | 0.170±.009 | 0.160±.006 |
| | LossVal (epochs=5) | *0.355*±.014 | 0.228±.009 | *0.292*±.010 | *0.292*±.007 |
| | LossVal (epochs=30) | **0.421**±.016 | *0.203*±.011 | 0.321±.011 | 0.315±.009 |
| **nomao** | AME | 0.063±.011 | 0.059±.010 | 0.065±.011 | 0.062±.006 |
| | DVRL | 0.309±.015 | 0.186±.007 | 0.246±.011 | 0.247±.007 |
| | Data Banzhaf | 0.195±.008 | 0.165±.009 | 0.183±.009 | 0.181±.005 |
| | Data-OOB | 0.364±.011 | 0.167±.006 | 0.272±.009 | 0.268±.007 |
| | Influence Subsample | 0.195±.008 | 0.178±.009 | 0.189±.008 | 0.187±.005 |
| | KNN-Shapley | *0.483*±.013 | 0.200±.009 | *0.302*±.009 | *0.328*±.009 |
| | LAVA | 0.051±.004 | **0.280**±.029 | 0.205±.020 | 0.179±.013 |
| | Leave-One-Out | 0.210±.010 | *0.193*±.010 | 0.202±.010 | 0.202±.006 |
| | LossVal (epochs=5) | 0.522±.014 | 0.163±.006 | 0.353±.011 | 0.346±.011 |
| | LossVal (epochs=30) | **0.637**±.014 | 0.124±.005 | **0.395**±.011 | **0.385**±.014 |
| **pol** | AME | 0.075±.013 | 0.006±.002 | 0.012±.005 | 0.031±.005 |
| | DVRL | 0.295±.011 | 0.170±.007 | 0.239±.009 | 0.235±.006 |
| | Data Banzhaf | 0.133±.013 | 0.029±.007 | 0.075±.012 | 0.079±.007 |
| | Data-OOB | 0.316±.011 | 0.174±.007 | 0.248±.009 | 0.246±.006 |
| | Influence Subsample | 0.134±.013 | 0.019±.006 | 0.065±.012 | 0.073±.007 |
| | KNN-Shapley | *0.319*±.011 | 0.176±.006 | 0.258±.010 | 0.251±.006 |
| | LAVA | 0.064±.004 | **0.686**±.016 | 0.400±.015 | *0.383*±.017 |
| | Leave-One-Out | 0.143±.014 | 0.027±.008 | 0.088±.013 | 0.086±.007 |
| | LossVal (epochs=5) | 0.555±.016 | *0.228*±.009 | *0.375*±.012 | 0.386±.011 |
| | LossVal (epochs=30) | **0.712**±.014 | 0.231±.009 | **0.420**±.012 | **0.454**±.013 |

Table 12: Average of the noisy sample detection F1 scores on the regression task, averaged over all noise rates. The number after $\pm$ indicates the standard error. Higher is better.

| | | Noisy Labels | Noisy Features | Mixed Noise | Overall Average |
|---|---|---|---|---|---|
| **cpu_activity** | AME | 0.002±.001 | 0.002±.001 | 0.002±.001 | 0.002±.000 |
| | DVRL | 0.240±.015 | 0.199±.011 | 0.227±.013 | 0.222±.008 |
| | Data Banzhaf | 0.002±.001 | 0.002±.001 | 0.002±.001 | 0.002±.000 |
| | Data-OOB | 0.002±.001 | 0.002±.001 | 0.002±.001 | 0.002±.000 |
| | Influence Subsample | 0.002±.001 | 0.002±.001 | 0.002±.001 | 0.002±.000 |
| | KNN-Shapley | *0.268*±.010 | *0.269*±.010 | *0.269*±.011 | *0.269*±.006 |
| | LAVA | 0.012±.002 | 0.029±.008 | 0.021±.003 | 0.021±.003 |
| | Leave-One-Out | 0.002±.001 | 0.002±.001 | 0.002±.001 | 0.002±.000 |
| | LossVal (epochs=5) | 0.470±.014 | **0.409**±.009 | 0.464±.011 | 0.448±.007 |
| | LossVal (epochs=30) | **0.638**±.013 | 0.367±.009 | **0.474**±.010 | **0.493**±.009 |
| **kin8nm** | AME | 0.001±.000 | 0.002±.001 | 0.002±.001 | 0.002±.000 |
| | DVRL | *0.217*±.013 | 0.208±.010 | 0.223±.011 | 0.216±.006 |
| | Data Banzhaf | 0.001±.000 | 0.002±.001 | 0.002±.001 | 0.002±.000 |
| | Data-OOB | 0.001±.000 | 0.002±.001 | 0.002±.001 | 0.002±.000 |
| | Influence Subsample | 0.001±.000 | 0.002±.001 | 0.002±.001 | 0.002±.000 |
| | KNN-Shapley | 0.001±.000 | 0.002±.001 | 0.003±.001 | 0.002±.000 |
| | LAVA | 0.186±.007 | **0.403**±.014 | 0.271±.010 | 0.287±.008 |
| | Leave-One-Out | 0.001±.000 | 0.002±.001 | 0.002±.001 | 0.002±.000 |
| | LossVal (epochs=5) | 0.292±.009 | *0.247*±.008 | *0.270*±.009 | *0.270*±.005 |
| | LossVal (epochs=30) | **0.409**±.012 | 0.315±.010 | **0.364**±.011 | **0.363**±.007 |
| **pumadyn32nh** | AME | 0.002±.001 | 0.002±.001 | 0.002±.001 | 0.002±.000 |
| | DVRL | *0.201*±.009 | *0.196*±.008 | 0.196±.008 | 0.198±.005 |
| | Data Banzhaf | 0.002±.001 | 0.002±.001 | 0.002±.001 | 0.002±.000 |
| | Data-OOB | 0.002±.001 | 0.002±.001 | 0.002±.001 | 0.002±.000 |
| | Influence Subsample | 0.002±.001 | 0.002±.001 | 0.002±.001 | 0.002±.000 |
| | KNN-Shapley | 0.002±.001 | 0.004±.003 | 0.006±.003 | 0.004±.001 |
| | LAVA | 0.193±.007 | **0.735**±.020 | **0.403**±.017 | **0.444**±.016 |
| | Leave-One-Out | 0.002±.001 | 0.002±.001 | 0.002±.001 | 0.002±.000 |
| | LossVal (epochs=5) | 0.203±.007 | 0.192±.007 | *0.199*±.007 | *0.198*±.004 |
| | LossVal (epochs=30) | **0.248**±.007 | 0.203±.007 | 0.225±.008 | 0.225±.004 |
| **superconductivity** | AME | 0.002±.000 | 0.001±.000 | 0.002±.001 | 0.001±.000 |
| | DVRL | *0.256*±.014 | 0.171±.010 | 0.207±.008 | 0.211±.007 |
| | Data Banzhaf | 0.002±.000 | 0.001±.000 | 0.002±.001 | 0.001±.000 |
| | Data-OOB | 0.002±.000 | 0.001±.000 | 0.002±.001 | 0.001±.000 |
| | Influence Subsample | 0.002±.000 | 0.001±.000 | 0.002±.001 | 0.001±.000 |
| | KNN-Shapley | 0.222±.009 | 0.227±.009 | *0.225*±.009 | *0.225*±.005 |
| | LAVA | 0.126±.006 | **0.652**±.014 | **0.437**±.010 | **0.405**±.014 |
| | Leave-One-Out | 0.002±.000 | 0.001±.000 | 0.002±.001 | 0.001±.000 |
| | LossVal (epochs=5) | 0.331±.010 | *0.177*±.006 | 0.251±.008 | 0.253±.006 |
| | LossVal (epochs=30) | **0.360**±.011 | 0.063±.003 | 0.220±.007 | 0.215±.008 |
| **wave_energy** | AME | 0.002±.001 | 0.002±.001 | 0.002±.001 | 0.002±.000 |
| | DVRL | *0.263*±.017 | 0.228±.012 | 0.224±.014 | 0.238±.008 |
| | Data Banzhaf | 0.002±.001 | 0.002±.001 | 0.002±.001 | 0.002±.000 |
| | Data-OOB | 0.002±.001 | 0.002±.001 | 0.002±.001 | 0.002±.000 |
| | Influence Subsample | 0.002±.001 | 0.002±.001 | 0.002±.001 | 0.002±.000 |
| | KNN-Shapley | 0.002±.001 | 0.002±.001 | 0.002±.001 | 0.002±.000 |
| | LAVA | 0.206±.008 | **0.436**±.027 | *0.256*±.010 | *0.299*±.011 |
| | Leave-One-Out | 0.002±.001 | 0.002±.001 | 0.002±.001 | 0.002±.000 |
| | LossVal (epochs=5) | 0.499±.015 | 0.419±.014 | 0.454±.014 | 0.458±.008 |
| | LossVal (epochs=30) | **0.660**±.008 | *0.401*±.015 | **0.517**±.015 | **0.526**±.010 |
| **white_wine** | AME | 0.002±.001 | 0.002±.001 | 0.002±.000 | 0.002±.000 |
| | DVRL | 0.306±.024 | *0.184*±.008 | 0.230±.012 | 0.240±.010 |
| | Data Banzhaf | 0.002±.001 | 0.002±.001 | 0.002±.000 | 0.002±.000 |
| | Data-OOB | 0.002±.001 | 0.002±.001 | 0.002±.000 | 0.002±.000 |
| | Influence Subsample | 0.002±.001 | 0.002±.001 | 0.002±.000 | 0.002±.000 |
| | KNN-Shapley | *0.428*±.015 | 0.162±.008 | 0.288±.010 | *0.292*±.009 |
| | LAVA | 0.037±.005 | **0.233**±.023 | 0.139±.015 | 0.136±.010 |
| | Leave-One-Out | 0.002±.001 | 0.002±.001 | 0.002±.000 | 0.002±.000 |
| | LossVal (epochs=5) | **0.486**±.015 | 0.202±.007 | **0.340**±.011 | **0.343**±.009 |
| | LossVal (epochs=30) | 0.466±.015 | 0.184±.007 | 0.321±.011 | 0.324±.009 |

Table 13: Average runtime on the *2dplanes* dataset (classification). Some overhead from the benchmark code around it should be expected, but should be comparable across all methods.

|  | 1000 Datapoints |
|---|---|
| AME | 8min 28s 983ms |
| Beta Shapley | 3min 30s 701ms |
| DVRL | 0min 33s 25ms |
| Data Banzhaf | 2min 07s 145ms |
| Data-OOB | 4min 05s 226ms |
| Data Shapley | 3min 16s 276ms |
| Influence Subsample | 2min 51s 927ms |
| KNN-Shapley | 0min 00s 160ms |
| LAVA | 0min 00s 104ms |
| Leave-One-Out | 4min 04s 954ms |
| LossVal (epochs=5) | 0min 01s 819ms |
| LossVal (epochs=30) | 0min 10s 617ms |

Table 14: Average runtime on the *20Newsgroups* and *CIFAR-10* datasets (classification).

|  | 20Newsgroups | CIFAR-10 |
|---|---|---|
| Base Training Time | 1h 4m 15s 658ms | 0h 28m 31s 859ms |
| Training Time with LossVal | 1h 4m 52s 415ms | 0h 32m 55s 315ms |
| LossVal Overhead | 0h 0m 36s 757ms | 0h 4m 23s 455ms |
| DataInf Overhead | 0h 2m 0s 28ms | 0h 6m 23s 241ms |
| LAVA | 0h 0m 35s 95ms | 0h 3m 10s 639ms |
| KNN-Shapley | 0h 0m 22s 254ms | 0h 13m 38s 224ms |

## I   Extended Related Work

We compare our methods to representative methods from the main branches of Data Valuation methods, as described in Section 2. We use them as strong baselines to show the general feasibility of our approach. However, there is a multitude of data valuation approaches that we left out in the comparison for feasibility, that are still worth mentioning here.

There is a range of approaches to extend Data Shapley, either to make it more efficient or to achieve better results on the benchmarks (Schoch et al., 2022; Pombal et al., 2023; Panda et al., 2024; Cai, 2024; Zheng et al., 2024).

None of them performs so much better than Data Shapley and Beta Shapley that we felt the need to include them in the benchmark. Other approaches try to employ other useful ideas from game theory, like the Banzhaf value (which is included in the comparison) and the Winter value (Chi et al., 2024).

Furthermore, there are some model-free approaches similar to LAVA(Just et al., 2023; Lin et al., 2024; Kessler et al., 2024) and approaches that apply data valuation without the use of a validation set (Jahagirdar et al., 2024). Other approaches apply methods similar to data valuation to machine learning models, datasets, data clusters, or distributions (Sun et al., 2024; Tarun et al., 2024; Xu et al., 2024; 2023; Yona et al., 2021). Instead of assigning a value to each data point, they assign a value to each model, dataset, data cluster, or distribution, respectively. Some modifications to data valuation allow a joint valuation of model and data point, or data points and data "cells" Sun et al. (2024).

Influence-based approaches, like Influence Functions, are generally quite inefficient. We used Influence Subsample to approximate the exact calculations of Influence Functions. Other influence-based approaches, such as Gradient Sketching (Schioppa, 2024) were left out.

Other notable methods left out in the comparison are Simfluence (Guu et al., 2023) and Neural Dynamic Data Valuation (NDDV) (Liang et al., 2024). Simfluence executes multiple training runs and trains a second model on the loss over time of a training run, based on the order in which the data points are sampled. The second model is used to simulate more training runs, which can then be used to estimate how important each data point is. NDDV uses optimal control strategies to understand the importance of data points without needing to retrain the Shapley utility function.

Aside from that, some interesting applications of data valuation are described in the literature. Nerini et al. (2024) study the applications of data valuation for graph-based data and data markets. An increasing number of papers focuses on the use of data valuation in an economic context or data markets (Tian et al., 2023; Agarwal et al., 2019; Chen et al., 2017; 2019; Li et al., 2015; Raskar et al., 2019; Mieth et al., 2024). Wang et al. (2024b) demonstrate data valuation methods can be attacked in an adversarial manner. Tian et al. (2024) study how data valuation can be made more robust to deletion of data points.

