# OpenReview forum: "LossVal: Data Valuation using Weighted Loss Functions"
_TMLR — Under review for TMLR_

### Review · Reviewer_pJB7 · 2026-06-19

**Summary Of Contributions:**

The paper proposes a data valuation method, where an auxiliary network $\omega$ and a common neural network $\theta$ are coupled in an optimization. The weight of the auxiliary network is regarded as the improtance score of each data sample.

**Additional Comments:**

Section 3 "LossVal" needs a more clear illustration and organization.

**Audience:**

Yes

**Audience Explanation:**

Data valuation/selection is valuable in both research community and industry.

**Claims And Evidence:**

No

**Claims Explanation:**

The pipeline and the motivation of the method is not clearly illustrated. Why do the authors design "LossVal"? What are the origin and the motivation of $\omega$? Please refer to the "Requested Changes".

**Requested Changes:**

I am not an expert of Data Valuation. I suggest the authors should re-organize the writing. New readers cannot get the motivation and the key design of LossVal.

1. Why can LossVal reflect the improtance of each data sample? It seems that $\omega_{i}$ is a coefficient of each sample. Why can $\omega$ represent the importance of each data sample? Is LossVal a loss function? Does optimization (e.g., Stochastic Gradient Descent) on LossVal exist? An illustration figure of the pipeline or a pseudocode is expected. A more detailed section 3 (i.e., "LossVal") is expected.

2. What is the scaling behavior of the method? What will happen if $N$ increases?

3. As for the term $\mathcal{L}_{\omega}(y, \hat{y})$ in LossVal, more illustrations of the term is essential. Why can the term represent the importance of the data sample under either CE and MSE cases?

4. As for the term $OT(X_{train}, X_{val})$ in LossVal, it seems measures the distance between the training dataset and the validation dataset, where something like "generalization" property is evaluated. So, what is the role of this term in LossVal?

---

> ### Author Response · Authors · 2026-07-08
> **Thank you for your review; we revised the manuscript based on your review and added it to OpenReview. Changes are marked in Blue/Red in the PDF.**
>
> We gratefully thank you for the time you invested in reviewing our manuscript. We hope to clarify the issues and questions raised below. Additionally, we revised our manuscript and updated it on OpenReview (changes in red).
>
> We think there might be a misunderstanding regarding the method: We did not train a complete auxiliary network. Instead, “we introduce instance-specific weights into the loss function”. Basically, we add learnable parameters to the MSE or cross-entropy loss. Each data point is weighted by its own weight, which gets optimized during gradient descent. After training, the point-wise weight is interpreted as the importance score.
>
> **Motivation**: The motivation for LossVal was to find a data valuation method that scales better with large models and datasets than previous approaches do. Instead of training a model thousands of times or calculating prohibitively expensive second-degree gradients, we use the information available during training. Our method can be easily plugged into the training process without requiring significant additional effort.
>
> **Requested changes**:
> Appendix A and B give a mathematical intuition for how LossVal finds useful importance scores. Yes, LossVal is a loss function (as stated on pp. 1, 5, and 6). Yes, we optimize the weights using stochastic gradient descent.
> Computational complexity is discussed in Section 7: When increasing N, the additional computation required for LossVal increases linearly. The complexity per batch increases quadratically for increased batch size.
> Because the MSE and cross entropy are incorporated into the LossVal loss function, as described on page 5 and in Appendix A.
>
> Thank you for your review. We revised the manuscript and hope that we clarified some of the ambiguities.

---

### Review · Reviewer_ysbh · 2026-06-25

**Summary Of Contributions:**

LossVal adds a learnable per-instance weight w inside the training loss. The loss is the product of a weighted target loss (weighted cross-entropy, Eq. 4, or weighted MSE, Eq. 5) and a squared weighted Sinkhorn optimal-transport distance between training and validation features (Eq. 3, Eq. 6). θ and w update jointly via gradient descent in one training run (Eqs. 1–2). Final or epoch-averaged w is the importance score.
The paper tests LossVal against 10 baselines (Data Shapley, Beta Shapley, LOO, KNN-Shapley, Data Banzhaf, AME, Influence Subsample, LAVA, DVRL, Data-OOB) on OpenDataVal (6 classification, 6 regression datasets); against KNN-Shapley, LAVA, and DataInf on CIFAR-10/ResNet-50 and 20Newsgroups/BERT; and in an active data-acquisition study on a crash-test dataset (1,122 NHTSA + 154 proprietary samples, Appendix D.3). It reports a loss-component ablation (Table 4), a complexity argument (O(n+T) vs. O(n·T) for LOO, O(n·p) for influence methods, Section 7), and a heuristic link to inverse-variance weighting under a no-shift assumption (Appendix B).

Strengths

- Targets a real cost bottleneck. Appendix H (Tables 13–14) shows faster runtime than retraining- and influence-based baselines.
- Wide evaluation: OpenDataVal suite, two large architectures, 10 baselines, one real-world case study.
- Multiply-vs.-add and squared-OT choices are ablated (Table 4) and derived (Appendix A), not just asserted.
- Reports unfavorable comparisons: LAVA on noisy-feature detection (Table 2); KNN-Shapley/DVRL on point addition (Figure 3).

Weaknesses

- In Table 8, LossVal's R² (0.131–0.137) is worse than the no-acquisition baseline (0.168) and random addition (0.160).
- LossVal trails LAVA on noisy-feature detection in both tasks, and trails KNN-Shapley in classification (Table 2).
- Section 5 gives two conflicting rankings for regression point removal.
- Appendix B's theory is scoped to linear regression/MSE, no shift. Authors state it is "neither likelihood-based nor equivalent to GLS or IRLS."
- LossVal's learning rate was tuned separately (Section 4); baselines used OpenDataVal defaults. Sinkhorn hyperparameters are not reported (Appendix F).
- Code/data are not yet available (Abstract).

**Audience:**

Yes

**Audience Explanation:**

- The loss-embedded weighting mechanism is distinct from the 23 prior methods listed alongside LossVal in Table 1.
- The single-training-run property addresses a cost barrier stated in Section 1, supported by runtime data (Appendix H).
- The negative findings are informative: strong on label-noise detection, weak on feature-noise detection (Table 2), not better than random in the acquisition study (Table 8).
- The crash-test application (Appendix C–E) broadens relevance to engineering domains with costly physical-experiment data.

**Broader Impact Concerns:**

A Broader Impact Statement is present. No extreme ethical risk is raised, but the statement is not calibrated to the paper's own results.

- The statement claims "significant utility in the Automotive Passive Safety domain" and a contribution to "safer roads." Table 8 shows LossVal does not beat random selection in the one acquisition experiment tested.
- Section 3 states that training points far from the validation distribution are down-weighted. In the crash-test setting (Appendix C–E), this could deprioritize rare but safety-critical configurations. The statement instead says "no specific societal consequences require immediate emphasis."
- The dataset includes 154 proprietary crash tests (Appendix D.3). The statement does not mention data governance or usage restrictions.

**Claims And Evidence:**

No

**Claims Explanation:**

Supported: the no-retraining/runtime claim (Appendix H); the no-performance-degradation result (Section 6, t(178)=-0.005, p=0.995; t(178)=1.350, p=0.179); "competitive" on CIFAR-10/20Newsgroups (Table 3).

Not adequately supported:

- "Robust across different types of noise" (Section 7). Table 2: classification F1 drops from 0.544 (labels) to 0.204 (features); regression from 0.464 to 0.256. LAVA beats LossVal on feature noise in both tasks.
- The acquisition case study (Section 1, Section 8) is framed as improving acquisition efficiency. Table 8 shows LossVal underperforms baseline and random selection. Section 5 itself notes "no strong differences between the data valuation methods."
- "Matches or outperforms state-of-the-art ... on both small and large datasets" (Section 7). LAVA wins noisy-feature detection (Table 2); KNN-Shapley/LAVA/DVRL win point addition (Figure 3, Table 10); KNN-Shapley/DVRL win point removal (Figure 4, Table 10).
- Section 5's two statements on regression point removal ("LossVal second-best" vs. "KNN-Shapley achieves much better results than all other methods") describe the same result and conflict.

**Requested Changes:**

Critical

- Reconcile repetition counts. Section 4: OpenDataVal experiments repeated 15 times. Appendix F.2: 25. Appendix E: acquisition procedure repeated 15 times. Appendix F.2: 50.
- Narrow or justify "robust across noise types" given the label/feature gap in Table 2.
- Revise the acquisition framing (Section 1, Section 8) to match Table 8, or add diagnostics for the gap with noisy-label detection performance.
- Clarify whether the tuned learning rate (Section 4) replaces η_θ, η_w, or both, relative to Table 7's baseline rates.
- Report Sinkhorn hyperparameters (regularization strength, iterations, tolerance) for OT_w (Eq. 6).
- Add significance testing for head-to-head comparisons in Tables 2, 3, 9–12, as already done in Section 6.
- State whether final-epoch or epoch-averaged w (Section 3) is used in each reported result.

Would strengthen the paper

- Explain why classification noisy-feature F1 drops slightly from epochs=5 (0.213) to epochs=30 (0.204) while noisy-label F1 rises sharply (Table 2).
- Extend Appendix B's intuition to cross-entropy/classification.
- Add a sensitivity analysis for η_w and the w_n=1 initialization.
- Add a second acquisition dataset or seed-level breakdown, given the repetition-count issue and the null result in Table 8.
- Relate the score-distribution differences in Figures 7–8 to LossVal's pattern of strengths and weaknesses across tasks.
- Clarify why DataInf appears only for CIFAR-10/20Newsgroups (Section 4), not in the main OpenDataVal comparison (Table 2).
- Standardize X_val (Eq. 3, Eq. 6) vs. X_test (Appendix A).

---

> ### Author Response · Authors · 2026-07-08
> **Thank you for your review; we revised the manuscript based on your review and added it to OpenReview. Changes are marked in Blue/Red in the PDF.**
>
> We gratefully thank you for the time you invested in reviewing our manuscript and for appreciating the extensive experiments we conducted. We hope to clarify the issues and questions raised below. Additionally, we revised our manuscript and updated it on OpenReview (changes in blue/red).
>
> **Answer to weaknesses**:
> Acquisition experiment: Unfortunately, none of the data valuation methods proved very helpful; every method falls within the standard error of random sampling. We updated the text to be more transparent and better reflect the results.
> LAVA on noisy feature task: We updated the text to better reflect the results.
> Conflicting rankings for regression point removal. There was a mistake in Table 10 that we have now corrected. We also fixed the textual description.
> LossVal's learning rate: LossVals learning rate tuning was necessary because it modifies the loss function. None of the other approaches impacted the training dynamics, so it was not necessary to separately tune the learning rate.
> Sinkhorn hyperparameters: We added them to the appendix hyperparameter list. We tried different hyperparameters, but the impact on performance was very small, so we settled on reasonable default values.
> Code and result data will be made available. Unfortunately, we cannot publish the crash test data. However, most of it is publicly available at the NHTSA database.
>
> **Answer to not adequately supported**:
> Robustness across noise: We improved the text to better reflect the results. LossVal is more robust to different noise types than the baselines, which excel at only one type each.
> Acquisition: See above
> We improved the text to better reflect the results
> We improved the description of the regression point removal
>
> **Requested changes**:
> Thank you for catching the inconsistencies! All experiments have been repeated 15 times.
> See above.
> The learning rate is always the same for η_θ and η_w to keep the setup simple and to account for the fact that tuning the learning rate separately may not be possible in a less controlled environment. The learning rate is tuned exclusively to maximize accuracy / minimize MSE.
> Added the Sinkhorn parameters to the appendix, see above.
> We are working on a good solution to include the significance tests for all experiments in a useful way, without taking up too much space. We will report back after adding them to the manuscript.
> Coefficients are taken after the last epoch. We made this clear in the manuscript now.
>
> **Would strengthen the paper**:
> Feature noise vs label noise: Feature noise is more nuanced than label noise. After training the model for longer, it might just learn to handle the noise better, thereby weakening the signal for the coefficients in the loss function. Though the difference is small and could also be due to random chance. We added this to the text.
> Missing DataInf: DataInf and Influence Subsample are both approximations of influence functions. Influence Subsample is less efficient and was too inefficient for the larger datasets; that’s why we only used it for the OpenDataVal benchmark. We do not expect both to perform very differently.
> X_val vs. X_test: Good catch! We fixed it.
>
> We also revised the impact statement to better align with the paper.
>
> Thank you very much for the very detailed review. It helped us immensely in improving the paper!

---

### Review · Reviewer_k77S · 2026-07-03

**Summary Of Contributions:**

The main idea of this paper is to identify influential training data points. There are several methods for this (influence functions, Shapley, attribution, integrated gradients). The paper proposes a new approach whereby the weights (importance scores) are optimized during training. The main contribution is defining a loss function that depends on the product of two terms: weighted loss function and a second term that measures alignment with validation set. Extensive experiments are presented to compare the method across several other baselines.

**Additional Comments:**

None

**Audience:**

Yes

**Audience Explanation:**

Identifying influential points is an essential problem in machine learning, and this adds a new method to the existing toolsets.

**Claims And Evidence:**

Yes

**Claims Explanation:**

To the best of my reading, the derivation of the gradient and the methodology is on a solid mathematical ground.

**Requested Changes:**

I would like to start by noting that the paper proposes an interesting idea, and the authors do an excellent job of putting the work in the context of existing works. The paper also does a good job of motivating LossVal, and I appreciate the details in Appendix about the experiments, which is the main strength of this paper. I think some parts of Appendix B could come after defining the LossVal loss. In the current format, the description after LossVal loss feels mechanical/routine, and does not really add much intuition (in contrast to Appendix B). I think the authors should re-consider this.

**Concerns and suggestions**

While the proposed LossVal framework is an interesting approach to data valuation, I have some concerns regarding its robustness and general applicability.

(1) My first concern is the validation set: The framework appears to depend heavily on the quality of the validation set, making the assigned data values susceptible to biases, limited sample sizes, or noisy validation data.

(2) My second concern is how OT might affect training stability: The optimization process that combines learnable data weights with Optimal Transport may introduce complex gradient dynamics that could affect training stability. The method may also face challenges when handling multimodal data distributions or extreme outliers.

(3) My third concern is dependence on training parameters: The learned data weights may vary across different training settings, such as changes in random seeds or batch sizes, raising questions about their consistency and reproducibility. Finally, because the approach relies on gradient-based optimization, its applicability is limited to differentiable models.

To strengthen the work, I encourage the authors to discuss these limitations more explicitly and, where possible, provide analyses or experiments that evaluate the framework's sensitivity to validation set quality, training configuration, and complex data distributions. It would also be valuable to discuss potential extensions that improve optimization stability and broaden the framework's applicability to a wider range of machine learning models. I do acknowledge some aspects of these have already been addressed (e.g., Noisy Sample Detection Tasks).

**Minor concerns**
1. as possible.This == > put space before This
2. instances, **i.,e**., all possible subset ===> Fix i.e.,
3. Possibly consider this method, and how it compares to the proposed method: https://arxiv.org/abs/1703.01365
4. Could you also discuss how the final complexity is obtained, given that the cost of computing OT is not trivial? I was not able to obtain the final complexity derivation.

---

> ### Author Response · Authors · 2026-07-08
> **Thank you for your review; we revised the manuscript based on your review and added it to OpenReview. Changes are marked in Blue/Red in the PDF.**
>
> We gratefully thank you for the time you invested in reviewing our manuscript and for appreciating the extensive experiments we conducted. We hope to clarify the issues and questions raised below. Additionally, we revised our manuscript and updated it on OpenReview (changes in red).
>
> Thank you for your suggestion regarding Appendix B. We improved the method description in the main paper by restructuring it and clarifying the method more.
>
> **Answer to concerns and suggestions**:
> Yes, that is absolutely right! Almost all approaches (including the baselines we used) rely on a clean validation set to assess the importance of a point or subset. There is ongoing research into data valuation without a validation set, e.,g., by doing k-fold cross-validation [1, 2]. We added another column to Table 1 to indicate which methods use a validation set (almost all). In the case of LossVal, using a validation set sampled from the same noisy distribution would mean that the optimal transport would see no difference between the validation and training sets. However, the weighted loss would still work, and the optimal transport would help regularize it.
> In Section 6, we compared the accuracy/MSE of training runs with and without LossVal. For each of the 6 classification and regression baselines, we trained 15 models with and without LossVal and found no statistically significant results. We did not find an experimental setting in which including LossVal significantly affected training stability, across tabular data, images, and text. The optimal transport tries to keep the training distribution similar to the validation distribution and appears to produce only a few outliers (see also Figure 7). It seems to act more like a regularization.
> We did our best to address this in the main experiments by running many different experimental settings, using different MLP hyperparameters for classification, regression, and crash (optimized for MLP without LossVal), and by repeating each run multiple times for each setting and dataset with different seeds. Table 3 includes additional experiments on text (MLP) and images (CNN). We also reported the standard error across repeated runs, which is similar to or lower than that of other data valuation approaches.
>
> We improved the discussion of the limitations and potential extensions. However, we think that thorough experimental and theoretical analysis of additional models (especially an extension for LLMs, which will need another modified loss function), multimodal data, and other extensions should be deferred to future work, given the limited space in the main paper and the already long appendix.
>
> **Answer to the minor concerns**:
> Thank you for spotting the mistakes!
>
> (3) The paper “Axiomatic Attribution for Deep Networks” uses the gradients of the model to identify important features and attribute how important each is. This is very interesting and similar to influence functions, which use the second derivative to identify important data points. Influence functions are discussed in the related work; they are prohibitively costly to compute, so we used the “InfluenceSubsample” and “DataInf” baselines to approximate them.
>
>
> Thank you for pointing that out! It really is too imprecise in the paper.
>
> In data valuation, we are mainly concerned about scaling the method with respect to the training set size [3].
>
> The complexity of LossVal is roughly O(T + e*(n/b)*(b*v)), where T is the complexity of a full MLP training, e is the number of epochs, n is the number of samples, b is the batch size, and v is the size of the validation set.
> The sinkhorn distance is roughly quadratic in the number of points per batch, O(N*J) for N training and J validation samples. But we apply it only to a batch at a time, so when scaling up the total number of training samples, the OT per batch stays the same; only the number of batches increases.
> So, if we increase the number of samples, the additional computational cost of optimal transport increases linearly. That's why we simplified it to O(T + n).
>
> Again, thank you for so thoroughly reviewing our manuscript; it benefits greatly from your comments.
>
> [1] Jahagirdar, H., Wang, J. T., & Jia, R. (2024). Data valuation in the absence of a reliable validation set. Transactions on Machine Learning Research.
> [2] He, Q., Zhang, M., Zhang, J., Yang, S., & Wang, C. (2023, September). K-fold cross-valuation for machine learning using shapley value. In International Conference on Artificial Neural Networks (pp. 50-61). Cham: Springer Nature Switzerland.
> [3] Hammoudeh, Z., & Lowd, D. (2024). Training data influence analysis and estimation: A survey. Machine Learning, 113(5), 2351-2403.
> [4] Just, H. A., Kang, F., Wang, J. T., Zeng, Y., Ko, M., Jin, M., & Jia, R. (2023). Lava: Data valuation without pre-specified learning algorithms. arXiv preprint arXiv:2305.00054.